

# Gravity loop integrands from the ultraviolet

**Alex Edison[1,2]⋆, Enrico Hermann[3]† Julio Parra-Martinez[1]‡ and Jaroslav Trnka[4]○**

**1** Mani L. Bhaumik Institute for Theoretical Physics,
UCLA Department of Physics and Astronomy, Los Angeles, CA 90095, USA
**2** Department of Physics and Astronomy, Uppsala University, 75108 Uppsala, Sweden
**3** SLAC National Accelerator Laboratory, Stanford University, Stanford, CA 94039, USA
**4** Center for Quantum Mathematics and Physics (QMAP),
Department of Physics, University of California, Davis, CA 95616, USA

⋆ alexander.edison@physics.uu.se, † eh10@stanford.edu,
‡ jparra@physics.ucla.edu, ○ trnka@ucdavis.edu

## Abstract

We demonstrate that loop integrands of (super-)gravity scattering amplitudes possess surprising properties in the ultraviolet (UV) region. In particular, we study the scaling of multi-particle unitarity cuts for asymptotically large momenta and expose an improved UV behavior of four-dimensional cuts through seven loops as compared to standard expectations. For $\mathcal{N} = 8$ supergravity, we show that the improved large momentum scaling combined with the behavior of the integrand under BCFW deformations of external kinematics *uniquely* fixes the loop integrands in a number of non-trivial cases. In the integrand construction, all scaling conditions are homogeneous. Therefore, the only required information about the amplitude is its vanishing at particular points in momentum space. This homogeneous construction gives indirect evidence for a new geometric picture for graviton amplitudes similar to the one found for planar $\mathcal{N} = 4$ super Yang-Mills theory. We also show how the behavior at infinity is related to the scaling of tree-level amplitudes under certain multi-line chiral shifts which can be used to construct new recursion relations.

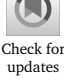

# 1 Introduction

The ultraviolet behavior of gravity scattering amplitudes has been of great interest for several decades [1–6]. Because of the dimensionful coupling constant, perturbative gravity is expected to develop ultraviolet (UV) divergences signaling the need for a UV completion. Indeed, it was found a long time ago that scattering amplitudes in Einstein gravity are UV divergent starting at two loops [2–4]. One well-known mechanism to improve and tame the UV behavior of a theory is to introduce supersymmetry which enforces certain cancelations of divergences in loop diagrams due to superpartners running in the loop. This famously leads to the cancelation of quadratic corrections to the Higgs mass but naively it can not solve the problem in gravity where power-counting would eventually win over any amount of supersymmetry.

This expectation is related to the standard picture where UV divergences of scattering amplitudes are closely linked to the appearance of counterterms which satisfy all symmetry requirements of a given theory. In this context, the existence of an $R^3$ counterterm in pure gravity is linked to the observed two-loop divergence. In contrast, supersymmetry forbids the $R^3$ term and increases the loop order at which the amplitude might diverge. For $\mathcal{N} = 8$ supergravity [7–9], the allowed counterterm consistent with all known symmetries of the theory has the form $D^8 R^4$ and implies a seven-loop divergence in four dimensions [10–18]. While there is an ongoing debate whether or not this is indeed the case, indirect evidence for the validity of the counterterm was given in [19] by calculating the five-loop UV divergence in the critical dimension, $D_c = 24/5$, implying that the standard argument holds [15][1].

On the other hand, recent results for $\mathcal{N} < 8$ supergravity [21] amplitudes suggest that our understanding of the relation between symmetries of gravity theories and their UV structure is not yet satisfactory [22–33]. Perhaps this is due to our incomplete grasp of supersymmetry itself and the lack of an off-shell superspace for higher amount of supersymmetry. However, if some of the amplitudes' observed properties can not be explained by supersymmetry or duality symmetries, it might point to new hidden symmetries or novel unexpected features of gravity.

For the time being, we would like to set aside the question of ultraviolet divergences in supergravity amplitudes. The aim of this paper, and more generally of the program initiated in [30, 34], is to use instead the gravity *loop integrand* as probe to explore the UV physics of

---

[1]Note that perturbative finiteness of $\mathcal{N} = 8$ SUGRA does not imply UV completeness [20].

gravity amplitudes, ask basic questions about analytic properties of gravitational scattering amplitudes, and eventually connect them to geometric ideas such as the Amplituhedron [35–38] for planar $\mathcal{N} = 4$ super Yang-Mills (SYM) theory and other positive geometries [35, 37, 39–42]. In this approach, we consider the ultraviolet region of amplitudes[2] as a broader concept. The UV properties are not just a binary statement about the presence or absence of divergences after integration, but more about the behavior of the S-matrix at infinite loop momenta. While unitarity implies the factorization of the S-matrix on infrared (IR) poles (at finite loop momenta), an analogous statement is not known for UV poles at infinite momenta –we denote those as *poles at infinity*.

Naively, power-counting predicts the degree of the pole at infinity for a given theory and should be manifest term-by-term in the expansion of amplitudes in a basis of Feynman integrals. This picture also acts behind the scene of most counterterm analyses, including the one for $\mathcal{N} = 8$ supergravity. Whenever we can identify a divergent integral in the expansion of the amplitude, we expect that this in turn reflects the divergence of the full amplitude. Any possible UV cancelations between terms that are *not* a consequence of gauge invariance or the known symmetries are therefore *unexpected* and directly point to some new property of the theory.

In [34], two of the authors pointed out that there are indeed cancelations which do improve the behavior of the loop integrand at infinity in comparison to the UV scaling of individual terms. While we observed this phenomenon in some isolated cases, in the present paper, we gather more comprehensive evidence and provide new results in this direction. Very importantly, we are going to show that the improved UV behavior of integrands is present only in $D = 4$ due to vanishing Gram determinants. This observation also explains the negative result in [30] and suggests that there are special features of four-dimensional gravity amplitudes still to be discovered.

Furthermore, we demonstrate that the improved scaling at infinity is a powerful constraint in the construction of supergravity amplitudes: in the generalized unitarity framework [43–46] it can be combined with the scaling of tree-level amplitudes under BCFW [47,48] deformations to fix loop amplitudes completely. All scaling constraints are *homogeneous conditions*, i.e. we do not match the amplitude functionally on cuts but rather demand that the unitarity based ansatz for the amplitude vanishes at certain points at infinity. The fact that homogeneous conditions are sufficient to uniquely fix gravity amplitudes also suggests a possible connection to the Amplituhedron geometry, in analogy to the discussion for $\mathcal{N} = 4$ SYM theory beyond the planar limit [49].

It is important to note that our discussion concerns the cuts of loop integrands. Based on unitarity, these cuts are given by products of tree-level amplitudes. Therefore, the behavior of loop integrands at infinite loop momenta is linked to large momentum shifts of trees. It has been known for a while that graviton tree-level amplitudes have a surprisingly tame large $z$ behavior for BCFW shifts [50–52] despite the naive power-counting expectations. This feature of gravity trees has been linked to improved UV properties of one-loop amplitudes in e.g. [53]. Here, we show that there are more general shifts of tree-level amplitudes with similar properties that can be used to reconstruct all graviton tree-level amplitudes.

The remainder of this work is structured as follows: In Section 2, we summarize salient features of the unitarity method and explain how basic UV properties of the diagrammatic expansion of amplitudes can be extracted from maximal cuts. In Subsections 2.3 and 2.4, we concretize the notion of a pole at infinity and potential cancelations thereof in the context of cuts. In section 3, we present one of the main results of our work. We analyze the scaling of multi-particle unitarity cuts for Yang-Mills and gravity in both general $D$ and $D = 4$. We

---

[2]We often use "amplitudes" synonymously with integrands of scattering amplitudes that still require integration over loop momenta.

find a surprising drop in the large momentum scaling in gravity when going to $D = 4$ which is attributed to the vanishing of a certain Gram determinant. In section 4 we lay out our second new result. We show, that the large momentum scaling behavior together with a few other homogeneous constraints are sufficient to uniquely fix the $\mathcal{N} = 8$ supergravity amplitude through three-loops and four external particles. In section 5, we attempt to understand some of the observed large momentum scaling improvements of gravity unitarity cuts in terms of properties of tree-level amplitudes under generalized shifts. We point out that under certain conditions, these new shifts lead to novel recursion relations of gravity tree-level amplitudes. We close in section 6 with some conclusion and an outlook to future work.

## 2 Integrands and cuts

The textbook formulation for the perturbative S-matrix is based on the expansion of scattering amplitudes in terms of Feynman diagrams. Higher order corrections in the perturbative series are encoded in loop amplitudes. For the $L$-loop $n$-particle amplitude in $D$ spacetime dimensions we can write,

$$\mathcal{A}_n^{L-\text{loop}} = \sum_{\text{FD}} \int d^D\ell_1 d^D\ell_2 \dots d^D\ell_L \, \mathcal{I}_n^{\text{FD}} \,, \tag{1}$$

where $\mathcal{I}_n^{\text{FD}}$ is a rational function of external momenta, loop momenta, polarization states, and possibly gauge theory data. The only poles in $\mathcal{I}_n^{\text{FD}}$ come from Feynman propagators and have the form $1/P^2$, where $P$ schematically represents a combination of external and loop momenta. Individual Feynman diagrams are not gauge invariant while the full amplitude $\mathcal{A}_n^{L-\text{loop}}$ is. We can decompose all Feynman diagrams into a basis of independent integrands (scalar integrals). The resulting decomposition of the amplitude is a linear combination of these basis elements with gauge invariant coefficients $c_k$,

$$\mathcal{A}_n^{L-\text{loop}} = \sum_k c_k \, I_k \quad \text{where} \quad I_k = \int d^D\ell_1 d^D\ell_2 \dots d^D\ell_L \, \mathcal{I}_k \,. \tag{2}$$

Searching for bases of loop integrands $\mathcal{I}_k$ is a very active area of research and many efficient methods have been developed in recent years to perform these calculations to higher multiplicities and higher loops in wide range of QFTs [54–58].

In the planar limit we can exchange the sum and the integration symbol and define the loop integrand $\mathcal{I}_n^{L-\text{loop}}$ as the sum of all contributing pieces prior to integration

$$\mathcal{A}_n^{L-\text{loop}} = \int d^D\ell_1 d^D\ell_2 \dots d^D\ell_L \, \mathcal{I}_n^{L-\text{loop}} \,. \tag{3}$$

It has been demonstrated in a number of cases that the loop integrand is not just an intermediate object in the calculation but rather it exhibits some remarkable properties deserving of an independent raison d'être. Prominent examples include new methods for constructing the planar $\mathcal{N} = 4$ SYM integrand using loop recursion relations [59], the connection to on-shell diagrams and Grassmannian [60], and the complete reformulation using the geometric Amplituhedron picture [35–38]. In contrast, there are a number of approaches advocating to calculate amplitudes directly without ever discussing integrands. These are based on bootstrap ideas of writing down appropriate function spaces for scattering amplitudes and imposing physical conditions to uniquely extract the scattering amplitudes, see e.g. [61–65].

## 2.1 Perturbative unitarity

Beyond the planar limit, the loop integrand can not be defined in the same way due to the lack of global variables[3]. Instead, we have to adhere to the diagrammatic expansion in Eq. (2). However, the loop integrand is still a very important concept which underlies the success of unitarity methods. Perturbative unitarity implies that the loop amplitude must factorize into lower-loop amplitudes when evaluated on *cuts*. In the most basic unitarity cut, two inverse propagators are set on-shell, $\ell^2 = (\ell + Q)^2 = 0$ and the amplitude factorizes into two pieces[4],

$$\underset{\substack{\ell^2=0 \\ (\ell+Q)^2=0}}{\mathrm{Cut}} \left[ \mathcal{A}_n^{L-\mathrm{loop}} \right] = \sum_{\substack{L=L_1+L_2+1 \\ \mathrm{states}}} \int d\mathrm{LIPS}_\ell \, \mathcal{A}_{n_1+2}^{L_1-\mathrm{loop}} \times \mathcal{A}_{n_2+2}^{L_2-\mathrm{loop}}$$

$$\underset{\substack{\ell^2=0 \\ (\ell+Q)^2=0}}{\mathrm{Cut}} \left[ \vcenter{\hbox{diagram}} \right] = \sum_{\substack{L=L_1+L_2+1 \\ \mathrm{states}}} \int d\mathrm{LIPS}_\ell \, \vcenter{\hbox{diagram}} \qquad (4)$$

where the sum is over the distribution of loop orders $L_1, L_2$ as well as the allowed on-shell states exchanged in the cut. The distribution of external legs $n_1, n_2$ of the subamplitudes have to be consistent with the cut channel $Q$ and are related to the number of external states $n$ via $n = n_1 + n_2$. The unitarity cut (4), and the basic tree-level factorization

$$\underset{Q^2=0}{\mathrm{Cut}} \left[ \mathcal{A}_n^{\mathrm{tree}} \right] = \sum_{\mathrm{states}} \mathcal{A}_{n_1+1}^{\mathrm{tree}} \times \mathcal{A}_{n_2+1}^{\mathrm{tree}}$$

$$\underset{Q^2=0}{\mathrm{Cut}} \left[ \vcenter{\hbox{diagram}} \right] = \sum_{\mathrm{states}} \vcenter{\hbox{diagram}} \qquad (5)$$

can be iterated to give rise to *generalized unitarity* [43–46]. In this setup, we can set to zero any number of propagators and the loop amplitude factorizes correspondingly[5].

$$\vcenter{\hbox{diagram}} \xrightarrow{\mathrm{Res}} \vcenter{\hbox{diagram}} \xrightarrow{\mathrm{Res}} \vcenter{\hbox{diagram}} \xrightarrow{\mathrm{Res}} \vcenter{\hbox{diagram}} \qquad (6)$$

This can be viewed as modifying the contour of integration to encircle poles (changing $R^{3,1}$ to involve $S^1$ around the poles), or equivalently, as taking residues of the loop integrand (see e.g. [69]). While the loop integrand $\mathcal{I}_n^{L-\mathrm{loop}}$ is not a unique rational function beyond the planar limit (due to the aforementioned lack of global variables), the unitarity cuts are still well-defined. In particular, the uniqueness and associated label problem is completely avoided if we consider situations where each loop is cut at least once and the residue is a product of tree-level amplitudes, as in (11) and (12).

The labels of the basis integrands $\mathcal{I}_k$ contributing to the expansion of the cut amplitude are unambiguously linked to on-shell legs in tree-level amplitudes. Importantly, we do not need to

---

[3]See [66, 67] for recent progress in that direction.

[4]In the following, we will drop the integration over the Lorentz invariant phase space $d\mathrm{LIPS}_\ell$.

[5]In massless theories in $D = 4$, the three particle amplitudes are special and completely fixed by Lorentz invariance. Momentum conservation and the on-shell conditions allow for MHV (blue vertex) and $\overline{\mathrm{MHV}}$ (white vertex) amplitudes, see e.g. [60, 68] for more details.

know the full amplitude beforehand in order to calculate unitarity cuts (As explained above, cuts are gauge invariant objects given by products of tree-level amplitudes.). There is no issue about basis choices, ambiguity of labelings or total derivatives etc. Knowing high multiplicity tree-level amplitudes suffices to calculate very high loop cuts, even if we do not have direct access to (uncut) amplitudes.

The unitarity cuts provide a considerable amount of information about the original loop integrand and indirectly about the loop amplitude. In *cut-constructible* theories [70] this information is complete, i.e. knowing all four-dimensional cuts allows us to uniquely reconstruct the loop integrand. In other cases, we have to include extra information. This can include soft or collinear limits, or knowing $D$-dimensional cuts [71]. Therefore, it is fair to say that cuts indeed specify the loop integrand uniquely, despite its explicit construction might be laborious and not practical for higher loops, e.g. due to the missing knowledge of the integrand basis.

The connection between properties of the loop integrand, its cuts, and the final amplitude is a very difficult question, but in certain cases we do have a partial or even complete understanding. In particular, the IR divergences of the amplitude come from very well-known regions of the loop integration, and are captured by soft and collinear cuts. In other words, any integrand which vanishes on these cuts must be IR finite and vice versa. Another peculiar feature is the uniform transcendentality property of certain integrals and $\mathcal{N} = 4$ SYM amplitudes: the integrals evaluate to polylogarithms of uniform degree. (For sufficiently complicated amplitudes and integrals, the space of polylogarithmic functions is insufficient, see e.g. [72–76]). This is closely related to logarithmic ($d \log$) singularities of the loop integrand and underlies much of the geometric story behind on-shell diagrams, the positive Grassmannian and the Amplituhedron. More practically, all these properties have been used to construct special integrands [49,77,78] that are relevant for deriving differential equations for families of Feynman integrals in canonical form [79–81]. On more general grounds, the cuts of loop integrands are related to the branch cuts of final amplitudes (for recent work in this direction for Feynman integrals, see e.g. [82,83]), despite a detailed link is not yet completely understood.

## 2.2 Cuts and UV

In the context of cuts, it is natural to ask how the UV behavior of amplitudes is encoded in loop integrands. On one hand, this has a simple answer: the UV divergences come from regions of large loop momenta. It is also relatively straightforward to determine the critical dimension $D_c$, i.e. the spacetime dimension where the first logarithmic divergence appears. This is done by rescaling the loop variables $\ell_k \to t \widetilde{\ell}_k$ and asking for what value of $D_c$ the integrand scales asymptotically like $dt/t$ as $t \to \infty$. As a trivial example, consider the scalar bubble integral at one loop. The $\ell \to t \widetilde{\ell}$ rescaling effectively corresponds to introducing a radial coordinate $t$. Transforming the measure $d^D\ell \to t^{D-1} dt\, d^{D-1}\widetilde{\ell}$ and neglecting the angular coordinates $\widetilde{\ell}$, we find

$$
I_b = \;\; \vcenter{\hbox{}} \;\; = \int \frac{d^D\ell}{\ell^2(\ell+p_1+p_2)^2} \xrightarrow{t \to \infty} \int \frac{dt}{t^{5-D}}, \tag{7}
$$

that the critical dimension for the bubble is $D_c = 4$. Said differently, fixing the spacetime dimension to $D = 4$, the bubble integral is logarithmically divergent, while scalar triangles and boxes are UV finite. This scaling analysis is exactly what is traditionally understood as power-counting loop momenta, and unless the remaining integral vanishes for auxiliary reasons, we learn everything about the presence of UV divergences of an integral from the large $t$ behavior.

Performing a similar analysis for the full amplitude is a bit more subtle. First, if we expand the amplitude in terms of Feynman diagrams (1) there could be cancelations between different diagrams as a consequence of gauge invariance. In order to account for such cancelations,

it is preferential to express the amplitude in terms of basis integrals with gauge invariant coefficients (2). In this case, barring any further surprises, it is expected that the UV behavior of the amplitude is given by the worst behaved integral.

This begs the immediate question: Is there an invariant way to determine the *minimal*[6] power-counting of an integral? The answer is that power-counting of individual integrals is dictated by the method of *maximal cuts* and thus by well defined, gauge invariant data of the theory itself. We consider a cut of the amplitude where the maximal number of propagators are set on-shell. This maximal cut singles out one contributing basis integral and its numerator must have the appropriate form to match the cut calculated as the product of tree-level amplitudes. Therefore, numerators for integrals with the maximal number of propagators (also called parent integrals) are fixed by maximal cuts. We can always add contact terms (shrinking propagators of parent integral) to a parent diagram and rotate the basis but this does not change the power-counting of the irreducible piece which is uniquely associated to the parent diagram and is required to match the maximal cut functionally.

One particular example to have in mind is an integral which contributes to the four-particle $\mathcal{N} = 8$ supergravity amplitude and will play a role in our later discussion. The maximal cut corresponding to this integral is

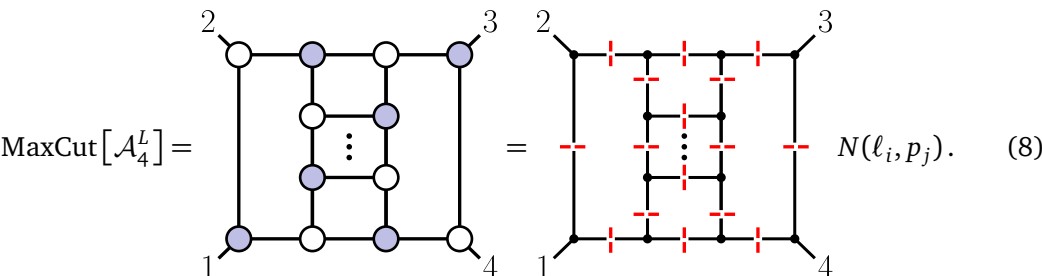

$$\text{MaxCut}\left[\mathcal{A}_4^L\right] = \quad = \quad N(\ell_i, p_j). \tag{8}$$

Matching the field theory cut of $\mathcal{N} = 8$ SUGRA on the left hand side of (8) requires the numerator of the local diagram to be $N(\ell_i, p_j) = (\ell_1 \cdot \ell_2)^{2(L-3)}$ modulo terms of lower power-counting in the $\ell_i$, or terms which vanish on this maximal cut (contact terms).

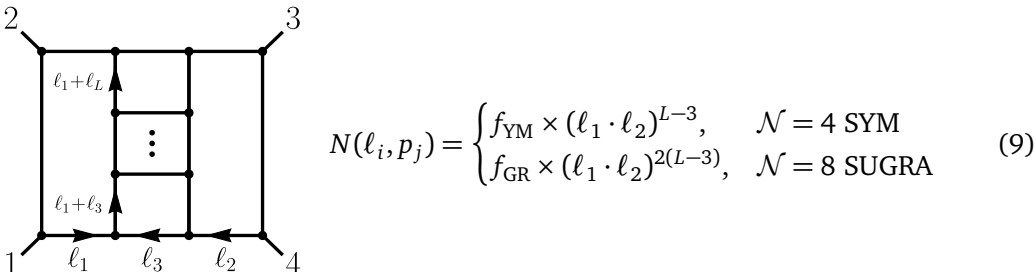

$$N(\ell_i, p_j) = \begin{cases} f_{\text{YM}} \times (\ell_1 \cdot \ell_2)^{L-3}, & \mathcal{N} = 4 \text{ SYM} \\ f_{\text{GR}} \times (\ell_1 \cdot \ell_2)^{2(L-3)}, & \mathcal{N} = 8 \text{ SUGRA} \end{cases} \tag{9}$$

In fact, (9) is a representative of the worst behaved diagram relevant for supergravity amplitudes in the UV. Continuing this line of logic, we see that this integral is divergent for $L \geq 7$ in four dimensions which suggests the presence of the $D^8 R^4$ counterterm in $\mathcal{N} = 8$ supergravity. If extrapolating the UV divergence of the full amplitude from the worst behaved local integral were legitimate, we would conclude that the amplitude indeed diverges starting at seven loops. Note that the power-counting of $\mathcal{N} = 4$ SYM is such that all diagrams stay UV finite to any loop order in $D = 4$.

---

[6]Roughly, "minimal power-counting" denotes numerator polynomials with the lowest possible degree in the loop variables $\ell_i$. For a detailed definition and various subtleties, see e.g. [84]. Note that one can always write a basis of integrands with higher power-counting that contains the minimal power-counting basis as a subspace. Superficially boosting the power-counting this way is not what we mean here.

There is an obvious caveat in the extrapolation argument: it is possible for UV divergences to cancel between various seven-loop diagrams, making the final amplitude UV finite (in $D = 4$). This would then result in a zero coefficient for the $D^8 R^4$ counterterm. A direct seven-loop calculation is not within current reach, but analogous $\mathcal{N} < 8$ calculations revealed that at lower loops there indeed occur *enhanced* cancelations of UV divergences between various terms making the result surprisingly finite [23, 24, 29, 30]. (A detailed discussion of the status of UV divergences in non-maximal supergravity theories, and various string and symmetry based analyses are beyond the scope of this work and a review can be found in the introduction sections of most of the references cited here.) On the other hand, the direct $\mathcal{N} = 8$ supergravity calculation for $L = 5$ in the critical (fractional) dimension $D_c = 24/5$ showed that there were no cancelations of this sort and the naive power-counting extrapolation was indeed the correct one [19]. Conservatively, this seems to seal the fate of the $D^8 R^4$ counterterm with an expected UV divergence at seven loops, assuming there is nothing special about $D = 4$ compared to the general $D$-dimensional gravity amplitudes. While we can not claim anything concrete about UV divergences in this work, we will show that four-dimensional gravity loop integrands indeed behave in a surprisingly good way.

## 2.3  Poles at infinity

Instead of a direct integration approach which faces technical challenges when attempting to go to seven loops, we take a different path to explore the physics of the UV structure of gravity. In particular, we focus on poles at infinity in the loop integrand evaluated on unitarity cuts. On one hand, studying the behavior of cuts does not directly tell us much about the UV divergences of the full amplitude as performing cuts effectively changes the contour of integration (see discussion in subsec. 2.1). On the other hand, we gain access to a richer set of statements about the behavior of the loop integrand at infinite loop momenta, beyond a binary statement about the presence or absence of a UV divergence. In particular, we are interested in the broader question of how physical principles constrain the behavior of the loop amplitude at infinity. As summarized in the beginning of Sec. 2, we know that unitarity dictates that the loop integrand factorizes when evaluated on the propagator poles. These factorization poles are in the IR (at finite momentum), but no analogous statements are known about the poles at infinite loop or external momenta. The behavior at infinity is also closely related to symmetries. In planar $\mathcal{N} = 4$ SYM for example, the (complete) absence of poles at infinity is a direct consequence of dual conformal symmetry [85, 86].

The aforementioned UV scaling $\ell \to t \widetilde{\ell}$ determines the presence (and degree) of UV divergences but probes infinity in a generic direction $\widetilde{\ell}$. As we will see later, there are special directions $\ell \to t \ell^*$ with $t \to \infty$ where the naive (power-counting) expectation does not work and the pole at infinity is absent (or has lower degree). These directions naturally appear on cut surfaces where the loop momentum gets partially fixed by on-shell conditions. Starting from the cut surface we subsequently send the loop momenta to infinity respecting the on-shell conditions.

$$\text{Cut}\left[ \vcenter{\hbox{\includegraphics{cut}}} \right] = \sum_{\text{states}} \vcenter{\hbox{\includegraphics{factorized}}} \longrightarrow \begin{array}{l} \ell_1^* = t \lambda_{\ell_1} \widetilde{\lambda}_{\ell_1} \\ \ell_2^* = t \lambda_{\ell_2} \widetilde{\lambda}_{\ell_2} \end{array} \qquad (10)$$

In fact, the necessity to first cut and then send the loop momentum to infinity is not optional and is forced on us if we want to discuss the behavior of the full loop integrand, not just individual basis integrals. This is because approaching the poles at infinity directly suffers from the same labeling problem described in subsec. 2.1: without cutting, $\ell$ means different

things in different diagrams, and therefore asking for a global meaning of $\ell \to \infty$ is ill-defined.

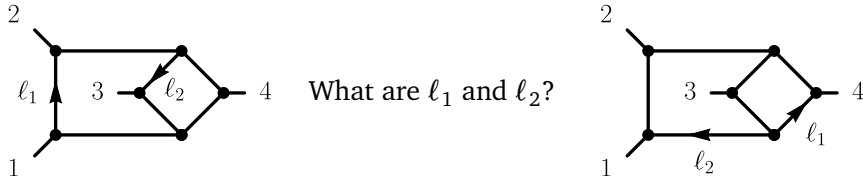

Figure 1: Ambiguity in labeling loop momenta in a given contribution to the integrand.

To be able to approach the UV limit in a well-defined manner we therefore have to first cut a certain number of propagators. In particular, we have to cut all loop momenta, at least in the minimal way, to factorize the loop integrand as a product of tree-level amplitudes. Then we can scale these cut loop momenta to infinity in various ways and ask how the integrand behaves under these scalings.

## 2.4 Cancelations

As we discussed before, one can compute the cut function as a product of tree amplitudes and perform the scaling limits explicitly without knowing the full integrand in the first place (by full integrand, we mean the knowledge of all coefficients $c_k$ in Eq. (2)). However, it is still very useful to compare a particular behavior at infinity of the (cut) loop integrand with the behavior of basis integrands ($\mathcal{I}_k$ in Eq. (2)) which contribute to the amplitude. The conservative expectation is that the scaling of the loop integrand on a particular pole at infinity is dictated by the basis integrands with the worst UV behavior (highest degree pole in the large $t$ limit). In the extreme case of maximal cuts this is indeed the case: only one basis integrand contributes and the behavior of the loop integrand is given by this term. In fact, this was used in subsection 2.2 to determine the power-counting.

If we cut fewer propagators, more basis integrands contribute, and there is a chance for cancelations. We initiated this work in [34] for various cuts in $D = 4$ and indeed found such cancelations where the loop integrand is better behaved at infinity than individual terms. While this initial study was very suggestive, it left some important questions unanswered: What is the role of $D = 4$ vs general $D$? Are the cancelations present only for special cuts? What are implications for the final amplitude?

We will answer the first two questions in this paper, while the third (most difficult) has to be relegated to future work. We know that the complete absence of poles at infinity leads to a simpler structure of integrated results. However, it is not clear how the absence of a particular pole at infinity is encoded in the final integrated answer.

We mainly focus on the most minimal cuts which specify unique labels and therefore allow us to talk about poles at infinity for the full (cut) loop integrand. From this perspective, the multi-particle unitarity cut is a prime representative,

$$F(\ell_k, p_j) = \underset{\ell_k^2=0}{\text{Cut}}\left[\mathcal{A}_n^{L-\text{loop}}\right] = \sum_{\text{states}} \mathcal{A}_{2+L+1}^{\text{tree}} \times \mathcal{A}_{2+L+1}^{\text{tree}}. \qquad (11)$$

The residue of the loop amplitude on this cut is given by the product of two tree-level amplitudes (integrated over the remaining phase space [see footnote 4] and summed over the

exchanged on-shell states). Our goal is to study the behavior of this cut in the UV region where the on-shell (cut) loop momenta $\ell_k$ approach infinity, $\ell_1, \dots, \ell_L, \ell_{L+1} \to \infty$, and compare the full cut to the contributing basis integrands,

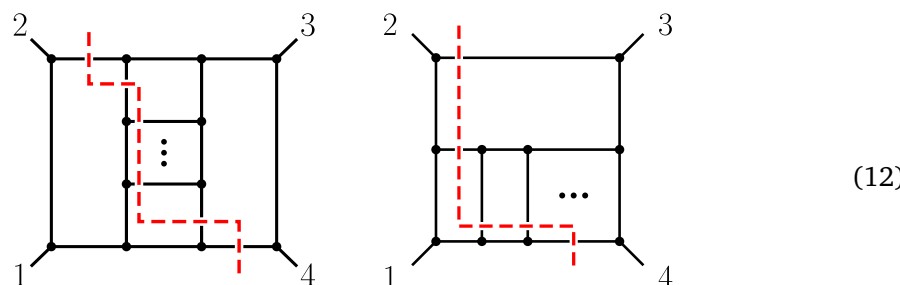

$$(12)$$

The only comparable analysis was done for certain two-loop four-point amplitudes [30], where it was concluded that the $D$-dimensional amplitude has the same scaling as the contributing integrals and no cancelations occur. In [34] the analysis was repeated in $D = 4$ finding an improved scaling at infinity of the cut amplitude compared to individual integrals. This signals that cancelations indeed happen in $D = 4$.

## 3 Improved scaling at infinity, general $D$ vs. $D = 4$

We first focus on the multi-particle unitarity cut illustrated in (11) for four-particle amplitudes in $D$ dimensions. All internal propagators visible in the figure are cut and impose $L+1$ on-shell conditions,

$$\ell_1^2 = \ell_2^2 = \cdots = \ell_L^2 = \ell_{L+1}^2 = 0 \quad \text{where} \quad \sum_{k=1}^{L+1} \ell_k = -(p_1 + p_2). \tag{13}$$

On support of these cuts, $F(\ell_k, p_j)$ is a $(D-1)L-1$ parametric function of on-shell momenta $\ell_k$ which satisfy momentum conservation as in Eq. (13). On this cut surface, there are numerous options how to scale the on-shell momenta $\ell_k$ to infinity. A very general way how to do this scaling is to perform a shift

$$\ell_k \to \ell_k + t\, q_k, \quad \text{where} \quad (\ell_k \cdot q_k) = q_k^2 = 0, \text{ and } \sum_k q_k = 0. \tag{14}$$

The conditions imposed on the $q_k$ guarantee momentum conservation and the on-shellness of the shifted momenta. Under this shift we get another on-shell function $F(\ell_k, q_k, p_j, t)$ which now depends not only on the original momenta $p_j, \ell_k$ but also the shift parameters $q_k$ and $t$. We approach infinity by scaling $t \to \infty$ keeping $q_k$ generic, and organize the result as a series in $t$,

$$\lim_{t \to \infty} F = t^m F_m + \mathcal{O}(t^{m-1}). \tag{15}$$

We are interested in the parameter $m$ which controls the leading behavior of the cut integrand at infinity. For general $q_k$, we indeed find that the behavior of the $\mathcal{N} = 8$ supergravity, as well as the pure gravity loop integrand, is controlled by the worst behaved local diagrams such as the one depicted in Fig. 9 for $L \geq 4$. This is absolutely expected as a drop in the exponent for general shift values $q_k$ would very likely indicate a decrease in power-counting and therefore, an increase of the critical dimension for the UV divergence. However, from the analysis of $\mathcal{N} = 8$ as well as pure gravity amplitudes we know that this can not be the case.

### 3.1 Special shift in $D$ dimensions

We choose to further specialize our shift (14) to the subspace defined by

$$(q_i \cdot q_j) = 0 \qquad \text{for all } i, j, \tag{16}$$

where the shifted propagators are all linear in $t$ for $t \to \infty$,

$$(\ell_i + \ell_j + Q)^2 \to (\ell_i + t\, q_i + \ell_j + t\, q_j + Q)^2 \sim \mathcal{O}(t), \tag{17}$$

since the quadratic terms in $t$ cancel. Performing the calculation explicitly for $\mathcal{N} = 8$ SUGRA and $\mathcal{N} = 4$ SYM, we see that the loop integrand scales like

$$F_{\text{SUGRA}} \sim \frac{1}{t^4}, \qquad F_{\text{SYM}} \sim \frac{1}{t^{L+2}}, \tag{18}$$

which is in agreement with the scaling of the worst behaved diagrams, and no cancelations occur. In fact, in order to perform these $D$-dimensional scaling analyses, we analyzed the results constructed in [87–89] and calculated the scaling from these integrand representations rather than gluing tree-level amplitudes together. The reason for doing so is to avoid technical complications involved with higher multiplicity $D$-dimensional tree-level amplitudes. An

Table 1: Scaling behavior of the $\mathcal{N} = 8$ SUGRA and $\mathcal{N} = 4$ SYM multi-particle unitarity cuts under the deformation defined by Eqs. (14) and (16) for the **D-dimensional cut integrands** up to four loops.

|  | $L = 2$ | $L = 3$ | $L = 4$ |
|---|---|---|---|
| SUGRA | $t^{-4}$ | $t^{-4}$ | $t^{-4}$ |
| SYM | $t^{-4}$ | $t^{-5}$ | $t^{-6}$ |

example diagram with the worst UV behavior under the specialized shift (14) (combined with the constraint (16)) is

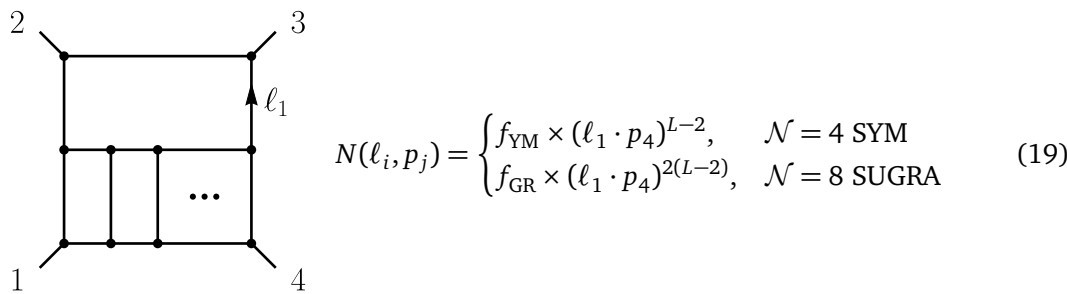

$$N(\ell_i, p_j) = \begin{cases} f_{\text{YM}} \times (\ell_1 \cdot p_4)^{L-2}, & \mathcal{N} = 4 \text{ SYM} \\ f_{\text{GR}} \times (\ell_1 \cdot p_4)^{2(L-2)}, & \mathcal{N} = 8 \text{ SUGRA} \end{cases} \tag{19}$$

On the multi-particle unitarity cut (11) of the diagrams in (12), there remain $3L + 1 - (L+1) = 2L$ uncut propagators, and the overall scaling of the diagram is

$$\text{SUGRA diagram scaling:} \quad \frac{(\ell \cdot p_4)^{2L-4}}{(\ell^2)^{2L}} \sim \frac{t^{2L-4}}{t^{2L}} \sim \frac{1}{t^4}, \tag{20}$$

independent of the loop order $L$. In comparison, we find that the diagram behaves like $\frac{1}{t^{L+2}}$ in $\mathcal{N} = 4$ SYM, which agrees with the scaling of the full loop amplitude.

## 3.2 Special shift in $D = 4$

Let us transition from the $D$-dimensional analysis to $D = 4$, where nontrivial cancelations in cuts with more on-shell propagators were previously identified in [34]. In going to $D = 4$, there is no change in the scaling behavior for individual basis integrand elements. To reiterate, the $\mathcal{N} = 8$ SUGRA basis elements scale like $1/t^4$, see (20), and the $\mathcal{N} = 4$ SYM basis elements fall off at infinity as $1/t^{L+2}$.

Having analyzed individual integrals, we now perform the calculation for the full amplitude. Instead of starting with the integrand in terms of local diagrams, we use four-dimensional Yang-Mills tree-level amplitudes calculated via BCFW (e.g. by the package of [90]) that are subsequently fed into the KLT relations [91–93] to obtain gravity trees. With this setup, we compute the UV scaling results through seven loops which are summarized in Fig. 2. We also obtain results for the non-supersymmetric theories and get e.g. $t^3$ for GR and $1/t^{L-2}$ for YM.

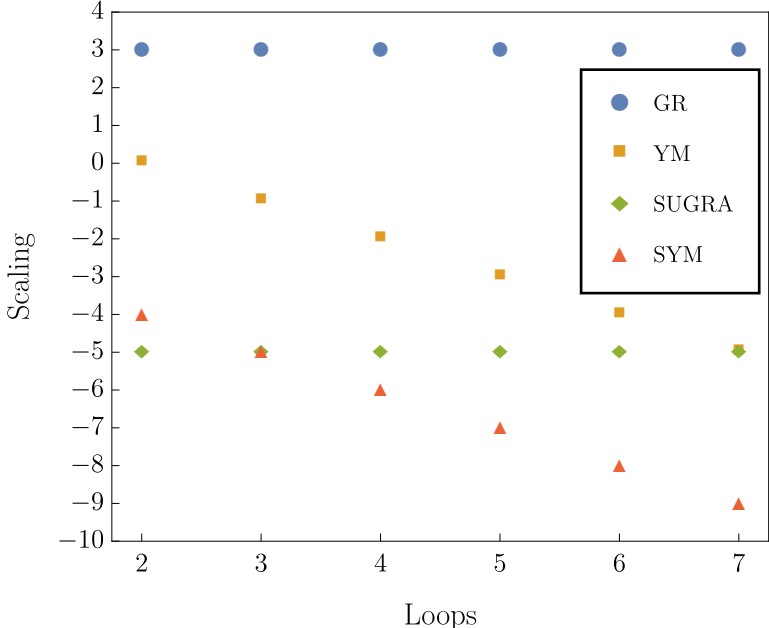

Figure 2: UV scaling of $\mathcal{N} = 8$ SUGRA, (planar) $\mathcal{N} = 4$ SYM, pure GR, and pure (planar)YM multi-particle unitarity cuts under **four-dimensional deformations** with results up to seven loops. The *Scaling* axis labels the leading $t$ behavior of the cuts as $t \to \infty$. The thin lines denote the scaling in D-dimensions, where the continuous part has been checked explicitly and the dashed part is conjectured. There is an overall improvement of one power in the large $t$ limit of gravity cuts with respect to $D$-dimensions; the same is not true for Yang-Mills.

While for (super) Yang-Mills theories there is no difference, and the $D = 4$ amplitudes scale identically as their general $D$-dimensional counterparts, in gravitational theories there is a **drop by one power**[7],

$$F_{\text{SUGRA}} \sim \frac{1}{t^5}, \quad F_{\text{GR}} \sim t^3 \qquad \text{for } 2 \leq L \leq 7. \tag{21}$$

Looking more closely at the $D$ to $D = 4$ transition, we find that (at least for $L = 2, 3$) the leading $1/t^4$ piece of the $\mathcal{N} = 8$ SUGRA amplitude has the following form

$$F_{\text{SUGRA}} \sim \frac{\Delta}{t^4} + \mathcal{O}\left(\frac{1}{t^5}\right), \quad \text{where} \quad \Delta = \left(\text{Gram}[q_1 q_2 \, p_1 p_2 p_3]\right)^2, \tag{22}$$

---

[7]Since Yang-Mills and gravity are closely related via KLT [91–93], it would be interesting to understand the drop in the large $t$ scaling of gravity multi-particle unitarity cuts in the $D{\to}D{=}4$ transition from this perspective.

and the $q_i$ in the Gram determinant denote the shift vectors of (14). Importantly, $\Delta$ *vanishes* in $D = 4$ thereby improving the UV scaling of the amplitude to $\mathcal{O}(\frac{1}{t^5})$. It is worth mentioning that at the loop orders at which we performed this analysis the power-counting of $\mathcal{N} = 8$ does not allow for a Gram determinant in the numerator of any single diagram. Crucially, many diagrams contribute to the cut and only the full sum assembles into the Gram determinant plus power suppressed terms at infinity. In higher loop cases, where Gram-determinants are allowed by power-counting, further (potentially badly behaved UV terms) drop out in strictly four spacetime dimensions. In our four-dimensional analysis of the cuts, any such drops are taken into account automatically by the use spinor-helicity variables. Even though, we have written out the explicit form of the Gram determinant only for the two- and three-loop integrands, this feature is clearly behind the cancellation of the leading power in the UV scaling of the integrand at higher loops. We conclude that there is a peculiar cancelation at infinity in gravity loop integrands on multi-unitarity cuts specifically in $D = 4$ owing to the special four-dimensional kinematics.

### 3.3 Comments

Studying the peculiar scaling properties of integrands at infinity begs the natural question about the meaning of this four-dimensional feature and what it can teach us about gravity amplitudes. We are far from having a complete answer and currently it is difficult to relate the improved large $t$ behavior of gravity cuts directly to new symmetries or implications for final amplitudes (including the status of UV divergences). However, several comments are in place.

**Shift in $D = 4$ and tree-level amplitudes**

Let us look at $D = 4$ more closely. We choose a particular shift of loop momenta $\ell_k \mapsto \widehat{\ell}_k$ which corresponds to a chiral shift, where the $\widetilde{\lambda}$ spinors are shifted proportional to a common reference spinor $\widetilde{\eta}$,

$$\widetilde{\lambda}_{\ell_k} \mapsto \widehat{\widetilde{\lambda}}_{\ell_k} = \widetilde{\lambda}_{\ell_k} + t\, z_k\, \widetilde{\eta} \quad \text{for} \quad k \in \{1, \dots, L+1\} \quad \text{subject to} \quad \sum_{k=1}^{L+1} z_k \lambda_{\ell_k} = 0, \qquad (23)$$

and the $\lambda_{\ell_k}$ remain unshifted[8]. We want to understand this behavior directly in the context of the tree-level amplitudes that enter the cut. In this case, Eq. (23) corresponds to a particular multi-line chiral shift where $n - 2$ legs of the tree are deformed,

$$\sum_{\text{states}} \quad \vcenter{\hbox{(diagram)}} \qquad (24)$$

The behavior of such deformed amplitudes for $t \to \infty$ depends on the helicities of the shifted (and unshifted) legs. The on-shell function representing the cut of the amplitude is a product of two tree-level amplitudes including the state sum over internal helicities. Therefore, the

---

[8]Note that a very similar shift has been discussed in the study of recursion relations for general 4D field theory tree-level amplitudes in Ref. [94]. In contrast to our loop-setup here, [94] shifted all external particles with such a chiral shift. We thank Henriette Elvang for insightful discussions.

individual tree-level amplitudes enter the expression in a particular correlated way (both helicities and shifted/unshifted momenta). For fixed internal helicities the product of two gravity tree-level amplitudes always scales as $t^3$ or better, while the individual tree-level amplitudes can scale up to $t^L$ at $L$ loops, their counterpart on the other side of the cut always compensates this poor scaling.

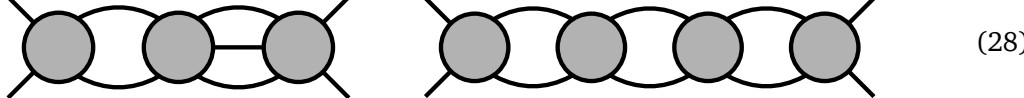

$$ \tag{25} $$

The existence of the improved behavior of tree-level amplitudes at infinity has been known for a very long time. The best example is the $1/t^2$ behavior of gravity tree amplitudes under BCFW shifts, which not only allows the reconstruction of the amplitudes from factorizations via the BCFW recursion relations, but it also implies the existence of bonus relations [51, 52, 95, 96]. For generic amplitudes that fall off at infinity sufficiently fast, such bonus relations can be recast as a sum rule on the residues of the amplitude at finite momenta

$$ \mathcal{A}_n(t) \sim \frac{1}{t^2} \quad \text{for} \quad t \to \infty \quad \longleftrightarrow \quad 0 = \oint_{\mathcal{C}_\infty} dt\, \mathcal{A}_n(t) = \sum_{\substack{i \in \text{poles} \\ \text{of } \mathcal{A}_n(t)}} \text{Res}_i\, \mathcal{A}_n(t = t_i). \tag{26} $$

In the supersymmetric case, our multi-line shift (23) is another example that leads to an improved behavior of deformed amplitudes at infinity (for appropriate helicity configurations) which allows for a number of bonus relations of the type (26). More general analyses are required to determine how the tree-level amplitudes behave at infinity for various shifts and what are the implications for loop integrands. In Refs. [94, 97–99] a number of shifts have already been considered and we add some new data points in Section 5.

**More cuts**

Besides the multi-particle unitarity cut described previously, there are several other cuts with a minimal number of on-shell propagators. In addition to permuting external legs in (11), we can also redistribute legs in the following way,

$$ \tag{27} $$

which also has higher-point generalizations where one considers all possible leg distributions on both sides. Apart from multi-particle unitarity cuts we can also discuss iterated versions thereof,

$$ \tag{28} $$

and study their behavior at infinity under the same chiral shifts. We indeed do find analogous drops in the large $t$ scaling specifically in $D = 4$ in all gravity theories similar to the original multi-particle unitarity cut discussed in subsection 3.2.

Together with the earlier analysis of higher cuts [34] it shows that these are not isolated findings and there must exist some systematic way to capture, explain and predict all these improved scalings in some unified way —predicting (rather than observing) which poles at infinity are absent, which are present and what is the degree.

# 4 Loop integrand reconstruction

To elaborate on the last point, we follow a particular path explored already in the case of $\mathcal{N} = 4$ SYM. We start with a general ansatz for the amplitude in terms of basis integrals and impose certain conditions trying to fix the amplitude uniquely. This ansatz procedure is at the heart of virtually all unitarity methods. In the most basic incarnation of generalized unitarity, the conditions correspond to matching field theory on a *spanning set of cuts*. In contrast, here we choose a very special set of constraints which is inspired by a possible geometric picture. All constraints must be **homogeneous** – meaning that we only impose vanishing conditions on the integrand ansatz, schematically

$$\mathcal{I}_{\text{ans}}\Big|_{\text{cond.}} = 0 \, , \tag{29}$$

as opposed to conventional unitarity, which matches the ansatz to non-zero functions via equations like

$$\text{Cut}[\mathcal{I}_{\text{ans}}] = \sum_{\text{states}} \mathcal{A}^{\text{tree}} \times \cdots \times \mathcal{A}^{\text{tree}} \, . \tag{30}$$

Below, we will list more specifically, the conditions utilized in the supergravity integrand construction up to three loops.

## 4.1 Homogeneous constraints

There are two conceptually distinct types of homogeneous constraints:

- Forbidden cuts: $1a$) field theory zeros, $1b$) helicity sector selection
- Theory specific: constraints specific to a given theory.

Forbidden cuts refer to cuts where field theory must be zero based on general principles, e.g. certain types of IR singularities never appear in amplitudes, or cuts vanish for specific helicity configurations. An example of a constraint from category $1a$) is a collinear cut where the loop momentum is proportional to an external momentum, $\ell = \alpha \, p_1$. Note that for gravitational theories, there are no collinear divergences [100, 101]. In the context of cuts, it has been shown in [68], that gravity integrands vanish in all collinear regions. In more general theories, such as Yang-Mills, this is not the case. In those theories, from an on-shell function perspective, it is easy to see that loop integrands factorize

$$\mathcal{I}_n^{L-\text{loop}} \to \frac{d\alpha}{\alpha(1-\alpha)} \times \widetilde{\mathcal{I}} \, , \tag{31}$$

where $\widetilde{\mathcal{I}}$ does not depend on $\alpha$. Therefore, the only poles in $\alpha$ are $\alpha = 0, 1$ which correspond to soft-collinear singularities making the momentum flow in propagators $\ell^2$ or $(\ell - p_1)^2$ zero. In the on-shell diagram language, the $\alpha$ parameter of Eq. (31) is associated to the face variable of the corresponding bubble on the external leg $p_1$,

$$\tag{32}$$

In contrast, individual integrals can have spurious collinear singularities not of the form (31) which must cancel

$$\Longleftrightarrow \quad \ell^\mu = \frac{Q_2^2}{2Q_2 \cdot p_1} p_1^\mu \, . \tag{33}$$

The cancellation can be used as an explicit constraint on an ansatz.

A simple example for a helicity-specific cut $1b$) appears in the context of quadruple cuts in $D = 4$. The relevant integral topologies for MHV one-loop amplitudes are two-mass-easy boxes. Solving the four on-shell conditions of the two-mass easy box integral gives two solutions. However, at the integrand level, MHV amplitudes only have nonvanishing residues on one of them, where the three-point corners are $\overline{\text{MHV}}$ amplitudes. This cut solution enforces collinearity conditions on the $\lambda$ spinors of the on-shell lines. On the second solution, the MHV loop integrand must vanish (due to $R$-charge or helicity counting) and therefore constitutes a forbidden cut.

$$
\begin{array}{c}
\xrightarrow[\ell_j^2=(\ell_j+p_j)^2=0]{\ell_i^2=(\ell_i+p_i)^2=0}
\end{array}
\quad
\begin{cases}
\lambda_{\ell_r} \sim \lambda_r, r \in \{i,j\} \quad \text{allowed,} \\[20pt]
\widetilde{\lambda}_{\ell_r} \sim \widetilde{\lambda}_r, r \in \{i,j\} \quad \text{forbidden.}
\end{cases}
\tag{34}
$$

In the context of planar $\mathcal{N} = 4$ SYM, there are two theory-specific constraints: the **absence of poles at infinity** and **logarithmic singularities**. The first constraint corresponds to the fact that the loop integrand never generates a singularity for $\ell \to \infty$ anywhere in the cut structure, i.e. there is never a pole (whether for real or complex $\ell$) which localizes $\ell \to \infty$. The latter constraint is more subtle and in momentum space it is only true for low $k$ amplitudes, where $k$ counts the helicity/$R$-charge of $\text{N}^{k-2}$ MHV amplitudes. In other cases, one can have elliptic and even more complicated singularities. However, if the amplitude is uplifted to bosonized momentum twistor variables, all singularities are logarithmic and near any pole $x = 0$ the loop integrand behaves as

$$
\mathcal{I} \xrightarrow{x=0} \frac{dx}{x} \quad \text{where } x = f(\ell, p).
\tag{35}
$$

Note that this property is much stronger than just having simple poles (which is automatic from Feynman propagators). The difference can be only seen on higher cuts, see [78] for a detailed discussion. For non-planar $\mathcal{N} = 4$ SYM amplitudes, the same properties were conjectured to hold [77], and verified in a number of cases. However, a general proof and deeper understanding of the theory-specific properties is still missing.

Both types of conditions alluded to above can be interpreted as the requirement that the loop integrand vanishes on certain cuts, schematically written as

$$
\text{Cut}_f \mathcal{A}_n = 0, \quad f \in \{\text{certain cuts}\}.
\tag{36}
$$

For planar $\mathcal{N} = 4$ SYM, the geometric picture for the loop integrand directly implies that the integrand function must be fully specified by these types of homogeneous conditions. This follows from the fact that a positive geometry in some positive variables $x_j$ can be defined by a set of homogeneous inequalities [35, 37, 41]

$$
h_a(x_j) \geq 0.
\tag{37}
$$

The differential form on that geometry (= loop integrand) can then be written as

$$
\Omega = \frac{N(x_j)}{\prod \text{poles}} \bigwedge_{j=1}^{4L} dx_j,
\tag{38}
$$

where the poles of $\Omega$ are dictated by the boundaries of the geometry. Because the numerator $N(x_j)$ is a polynomial in $x_j$, it is fully specified by its zeroes $x_j^*$. Geometrically, these zeroes correspond to special points *outside* the space defined by the inequalities in Eq. (37). Potentially, the denominator in (38) can generate singularities at locations $x_j^*$ where the inequalities (37) are violated. In order to not generate a spurious singularity, the role of the numerator is to put a zero at the location $x_j^*$. The crucial non-trivial statement is that in momentum space, these $x_j^*$ correspond exactly to the points $f$ of the vanishing cuts in Eq. (36) – the denominator structure of the loop integrand does in principle support such singularities, but the numerators must vanish in order to prevent the appearance of a pole. This is just a heuristic picture, which can be made more concrete in the context of the planar $\mathcal{N} = 4$ SYM [102]. There, even the numerator of the form (38) happens to be positive inside the positive geometry domain suggesting a dual Amplituhedron interpretation in which the the differential form is replaced by a volume integral.

We will not speculate further on the existence of a geometric picture for gravity amplitudes (nonetheless, it serves as ample motivation), but will instead investigate the ability to fully determine gravity amplitdues imposing *only* vanishing cuts (29), (36) on an ansatz.

## 4.2 Amplitude reconstruction

In this subsection, we focus on $\mathcal{N} = 8$ SUGRA as the simplest representative of gravitational theories, which is the most likely candidate to be fully fixed by homogeneous constraints. In particular, we make use of the following theory specific constraints: **improved behavior at infinity of cuts** discussed in Sec. 3, and **improved scaling of cuts under BCFW deformations** of external momenta. We begin by constructing the two- and three-loop four-point integrands of $\mathcal{N} = 8$ SUGRA.

**Two-loop four-point**

We first reconstruct the integrand of the two-loop four-point amplitude from the scaling constraints at infinity. Originally, the integrand was calculated in [87] in terms of a diagrammatic expansion along the lines of Eq. (2)

$$
\begin{array}{c} \phantom{x} \end{array} = s_{12}s_{23}s_{13}\mathcal{A}_4^{\text{tree}} \sum_{\sigma \in S_4} \left[ \frac{1}{4} \phantom{xxxx} + \frac{1}{4} \phantom{xxxx} \right], \quad (39)
$$

with the following numerators associated to each graph

$$
N \left[ \phantom{xxxx} \right] = N \left[ \phantom{xxxx} \right] = \left( p_{\sigma_1} + p_{\sigma_2} \right)^4 = s_{\sigma_1 \sigma_2}^2. \quad (40)
$$

In [34] two of the authors showed that this representation satisfies the improved scaling behavior at infinity when evaluated on the three-particle cut (11). Instead of merely observing the consistency of the numerators (40) with the UV scaling, we now demonstrate that the improved scaling conditions of Eq. (21) are sufficient to select these numerators from an ansatz.

The two-loop ansatz is built on the integral topologies shown in Fig. 3. For each integral, we write an ansatz for its numerator with the following properties

- We assign an overall factor of $s_{12}s_{23}s_{13}\mathcal{A}_4^{\text{tree}}$ to each diagram.

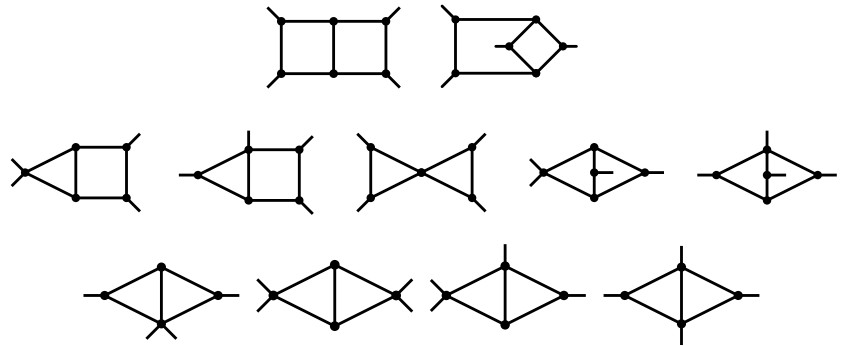

Figure 3: The integral topologies appearing in our ansatz for the two-loop four-point $\mathcal{N} = 8$ SUGRA amplitude.

- We allow all terms that can be fixed purely from the maximally-supported cut of the diagram. All contact terms are treated as separate topologies with their own degrees of freedom.
- We impose *triangle power-counting*: we only allow numerators that are equivalent to scalar triangles. In particular we do *not* allow terms of the form $(\ell_i \cdot p)(\ell_i \cdot q)$, see e.g. [58] for more details. Note that this is a very conservative assumption as triangle power-counting is worse than what is eventually necessary for the $\mathcal{N} = 8$ SUGRA examples discussed here.
- We impose diagram symmetry, that is, invariance of the numerator under all automorphisms of the skeleton graph.

In simple cases, the numerators are composed of $s_{ij}$ and irreducible numerators, see e.g. [103]. For more complicated diagrams, the requirement that the numerators obey all diagram symmetries can force the inclusion of *reducible numerators* whose coefficients are however completely locked to coefficients of irreducible ones. As such, they can be fixed on maximal cuts.

The two-loop planar double box for example carries a numerator ansatz which is a degree-two polynomial built from the following scalar product building blocks

$$\text{[diagram]} \longleftrightarrow s_{12}s_{23}s_{13}\mathcal{A}_4^{\text{tree}} \times \{s_{12}, \, s_{23}, \, p_1{\cdot}\ell_1, \, p_1{\cdot}\ell_2, \, p_2{\cdot}\ell_1, \, p_2{\cdot}\ell_2, \, p_3{\cdot}\ell_1\}^2, \quad (41)$$

where we have implicitly used momentum conservation to remove dependence on $p_4$. Note that this application of momentum conservation, as well as the need for diagram symmetries, has introduced *reducible* scalar products even in this simple case. We might expect such a numerator ansatz to have 49 free parameters. However, imposing the symmetries and triangle power-counting reduces the actual degrees of freedom to 6, all of which can be in principle fixed on the maximal cut of the diagram.

$$N\left[\text{[diagram]}\right] = s_{12}s_{23}s_{13}\mathcal{A}_4^{\text{tree}} \times \left[c_1\, n_1 + c_2\, n_2 + \cdots + c_6\, n_6\right]. \quad (42)$$

The individual numerator basis elements $n_i$ can be chosen as

$$
\begin{aligned}
n_1 &= s_{12}^2, \quad n_2 = s_{12}s_{23}, \quad n_3 = s_{23}^2, \\
n_4 &= s_{12}[\ell_1 \cdot (p_4 - p_3) + \ell_2 \cdot (p_1 - p_2)], \\
n_5 &= s_{23}[\ell_1 \cdot (p_4 - p_3) + \ell_2 \cdot (p_1 - p_2)], \\
n_6 &= [\ell_1 \cdot (p_4 - p_3)][\ell_2 \cdot (p_1 - p_2)].
\end{aligned}
\tag{43}
$$

Note that the basis numerators written in Eq. (43) explicitly depend on $p_4$ for compactness. Using momentum conservation, however, we can reduce all dot products to the basis elements introduced in Eq. (41). The remaining diagram numerators for the rest of the potential topologies in Fig. 3 are constructed in a similar manner. Specifically, the other diagram in the first row is also built as a degree two polynomial in the momentum products, while the second row each carries a degree one polynomial, and the final row diagrams are given undetermined rational coefficients. This ansatz contains the known integrand (40) by zeroing all free parameters except $c_1$ and its counterpart in the non-planar ladder, which are set to 1.

Next, we impose the homogeneous constraints on the ansatz constructed as above. The construction of four-point integrands does not require the use of forbidden cuts to project onto the desired helicity sector. Thus, we can solely focus on the homogeneous UV scaling conditions. We begin by requiring the appropriate behavior at infinity on the multi-particle unitarity cut kinematics (14), (16). Concretely, after calculating the cut of the ansatz, we shift the loop momenta via (23) to get a function that parameterizes the cut in terms of $t$

$$
\mathrm{Cut}_{\,\substack{\\ \bigcirc\!\bigcirc}}[\mathcal{I}_{\mathrm{ans}}] \xrightarrow{\text{chiral shift}} F(\{\ell, p\}, t),
\tag{44}
$$

which can then be series expanded in the limit $t \to \infty$ (15). In general this expansion of the ansatz will yield a Laurent series in $t$

$$
\lim_{t \to \infty} F(\{\ell, p\}, t) = \sum_{i=-\infty}^{\infty} F_i(\{\ell, p\}) t^i.
\tag{45}
$$

We then impose the observed scaling discussed in section 3.2. Specifically, we require that

$$
F_i(\{\ell, p\}) = 0 \quad \forall\, i > -5
\tag{46}
$$

for generic values of $\{\ell, p\}$, from which we extract constraints on the free parameters of the ansatz. For the rest of this paper, we will use shorthand of the form

$$
\propto \frac{1}{t^5}.
\tag{47}
$$

to denote this process of fixing parameters using cut scaling constraints. Enforcing this homogeneous condition determines the entire two-loop ansatz in terms of one parameter, except for the "kissing triangles" topology at the center of the second row in Fig. 3. Further consideration reveals that this is because no permutation of such integral topology contributes to the multi-particle cut. To resolve the missing information, we consider an "iterated" two-particle cut, where we impose scaling in one of the one-loop subdiagrams

$$
\propto \frac{1}{t^4}.
\tag{48}
$$

Notably, the scaling is slightly different for the one-loop subdiagram. While we do not discuss this feature here, it is covered in detail in the previous paper [34]. Imposing the iterated scaling fixes the numerator for the "kissing triangles". Thus, just imposing the particular behavior at infinity, we single out the known representation of the two-loop four-point $\mathcal{N} = 8$ supergravity integrand in Eqns. (39) and (40).

**Three-loop four-point**

At three-loop four-point, the combinatorics of the ansatz is much more involved. After careful counting, we are left with 2758 parameters[9] in 83 diagrams. As hinted at in the two-loop construction, to make sure all diagrams in our basis are constrained we need to consider the scaling on a spanning set of cuts. Specifically, we need to consider diagrams with different distributions of external legs as shown in Fig. 4a. Additionally, similar to the two-loop case, we need to consider the iterated cuts in Fig. 4b to constrain the factorizable integrals.

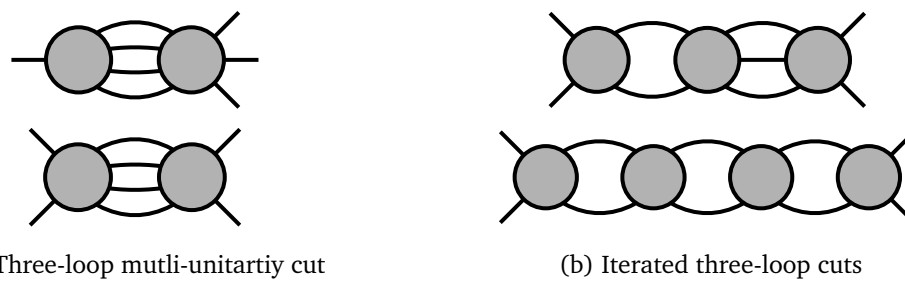

(a) Three-loop mutli-unitartiy cut          (b) Iterated three-loop cuts

Figure 4: Cut topologies considered in the UV construction of the three-loop four-point amplitude in $\mathcal{N} = 8$ SUGRA.

Imposing the appropriate scaling for the cuts of Fig. 4, we fix nearly all of the terms in the ansatz. However, there is a class of terms that the cut scalings cannot differentiate. For example, if we consider the ladder diagram, we see that the ansatz for its numerator is reduced to

$$
N\left[\begin{array}{c} {}^2 \qquad {}^3 \\ \\ {}_1 \qquad {}_4 \end{array}\right] = a_1 s_{12}^4 + a_2 s_{12}^3 s_{14} + a_2 s_{12}^2 s_{14}^2 \,. \tag{49}
$$

Each of these terms (and similar terms in other diagrams) scales as $t^{-6}$ on the multi-particle unitarity cut, and thus we require further homogeneous conditions to fix them.

A natural choice for the additional constraint is to impose the appropriate behavior of the multi-particle unitarity cut under BCFW shifts of external momenta. In fact, we know that the on-shell function corresponding to the multi-particle cut behaves like

$$
t\ \begin{array}{c}\\ \end{array} \Rightarrow F(t) \sim \frac{1}{t^2} \quad \text{for } t \to \infty \,, \tag{50}
$$

as a consequence of the behavior of the contributing tree-level amplitude. While all previous cuts do contain information about the behavior at infinity of cut (on-shell) loop momenta, the last condition also imposes constraints on the behavior at infinity for external momenta. If

---

[9]Note that the number of parameters quoted here is even more conservative and includes a few degrees of freedom that are beyond triangle power-counting. In particular, we did not remove all terms of the form $(\ell \cdot p)(\ell \cdot q)$ from sub-boxes as we did in the two-loop analysis (see discussion below (41)).

we consider the union of all these constraints, 2757 of 2758 parameters get fixed, leaving us only with one overall constant. Thus, the scaling conditions are sufficient to fully specify the amplitude without needing to compare with any specific values on a cut.

The above constructions show that four-particle amplitudes in $\mathcal{N} = 8$ SUGRA up to three loops can be fully defined by homogeneous conditions at infinity alone.

**One-loop $n$-point MHV**

A very interesting further application of the improved UV behavior is the reconstruction of the one-loop $n$-point MHV amplitude in $\mathcal{N} = 8$ SUGRA [93]. In the standard unitarity methods the result is given by the sum of box integrals where the coefficients correspond to leading singularities on the quadruple cuts [45]. In fact, the loop integrand also includes parity odd pentagons which integrate to zero but are needed to match all cuts properly. A very convenient set of basis integrals are chiral boxes [104].

The fact that there are no triangle and bubble integrals at one loop follows directly from the absence of poles at infinity as was shown in [53]. In contrast, our construction starts with a complete basis ansatz for the one-loop amplitude given by box and triangle integrals with numerators restricted by triangle power-counting. In this triangle-power-counting basis, the parity-odd pentagons become redundant [58]. Having set up the integrand basis, we are in the position to impose the UV scaling constraints on this ansatz. The absence of poles at infinity on all triple cuts links the chiral box and scalar triangle numerators together. In fact, looking more closely at the behavior of the two particle cuts,

$$F(\ell) = \hspace{3cm} \tag{51}$$

the behavior at infinity is actually even stronger than just the absence of a pole, see the discussion in [34] for more details. In fact, this is true for any direction $\ell \to t\ell$ at infinity on the unitarity cut,

$$F(\ell) \sim \mathcal{O}\left(\frac{1}{t^3}\right). \tag{52}$$

The mere absence of poles at infinity only requires the cut integrand to fall off like $\mathcal{O}\left(\frac{1}{t^2}\right)$ as $t \to \infty$. The improved behavior in Eq. (52) can in principle be used as a constraint, but does not add independent information in our one-loop MHV construction. Imposing this constraint might be necessary to construct $N^{k-2>0}$MHV amplitudes.

As opposed to the higher loop four-point examples, imposing the vanishing of the MHV amplitude on forbidden cuts is essential. This is completely natural, for an $n$-point amplitude where one is required to specify which of the $N^{k-2}$MHV sectors one is interested in. This is easily done by demanding that our ansatz vanishes on the non-MHV cut solution of the quadruple cut (34). Collecting all constraints from the unitarity cuts described above for all distribution of external legs in (51) we indeed fix all coefficients in the ansatz up to an overall factor.

## 4.3 (Non)-cut constructibility of $\mathcal{N} = 8$ amplitudes

As we saw in the previous subsection, the UV constraints are very powerful and sufficient to completely fix the loop integrand up to three loops at four point as well as for any number of points at one loop. While this is not an efficient way to construct amplitudes (other

methods, such as the Bern-Carrasco-Johansson double copy between gauge theory and gravity integrands [105,106] and its generalization [107], are much more efficient), it shows that the $\mathcal{N}=8$ supergravity loop integrand can be fully fixed (up to the orders checked and conjecturally more generally as well) using only homogeneous constraints at infinity. Note that we needed more than just the behavior on the multi-particle unitarity cut (or the iterated versions of that), but also the behavior of the cut integrand under BCFW shifts of external momenta.

This fact is not too surprising and for higher loops we need even more constraints at infinity. Due to gravity power-counting, we can easily see that at a sufficiently high loop order, even at four points, there are potentially diagrams with the same power-counting as parent integrals, that have no propagators in one of the loops at all[10]. Therefore, all multi-particle unitarity cuts (and in fact any cut that involves this loop) of this diagram vanish, and the amplitude is not cut-constructible.

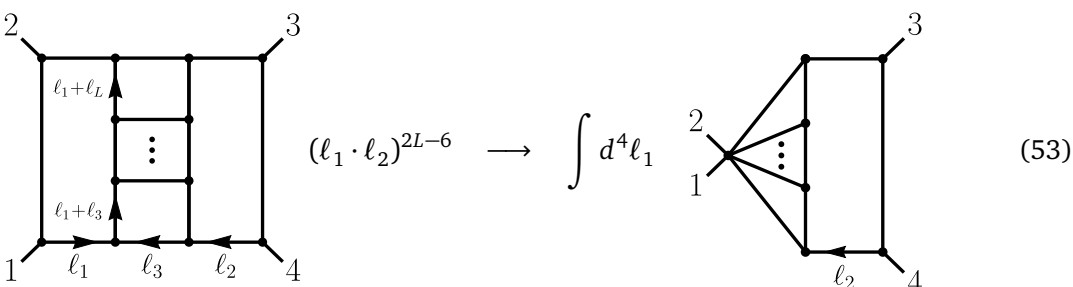

$$(53)$$

To illustrate this point, consider the numerator factor $(\ell_1 \cdot \ell_2)$ in the left diagram of (53). This term has the same asymptotic UV behavior as inverse propagators $(\ell_1 - p_1)^2$, $\ell_1^2$, $(\ell_1 + \ell_3)^2$, and so on. Therefore, starting at $L+1$ insertions of the numerator $(\ell_1 \cdot \ell_2)$, we can simultaneously write down terms that completely collapse all propagator factors as depicted in the right diagram of (53). This possibility first arises when $2L-6 = L+1$, i.e. for $L \geq 7$. This statement is based on completeness properties of integrand bases [58] for a given power-counting. When considering the correct numerator of the parent diagram on the left of (53) as dictated by the maximal cut, we also have to take into account the reduced diagram on the right of (53) in our ansatz. At seven loops, the numerator ansatz includes many terms such as

$$N \subset \{(\ell_1 \cdot \ell_2)^8, \ell_1^2(\ell_1 - p_1)^2(\ell_1 - p_{12})^2(\ell_1 + \ell_3)^2(\ell_1 + \ell_4)^2(\ell_1 + \ell_5)^2(\ell_1 + \ell_6)^2(\ell_1 + \ell_7)^2, \ldots\}, \qquad (54)$$

where the second term represents the collapsed integral on the right of figure (53). This diagram does not have any propagators in $\ell_1$ and therefore vanishes on all unitarity cuts. Note that kinematically there is no way to forbid such terms as they can freely mix with numerators that have non-zero contributions to cuts. Unless there is some extra constraint or mechanism which protects such integrals to appear, we have to conclude that the $\mathcal{N}=8$ supergravity integrands are (for sufficiently high $L$) *not cut-constructible* and further conditions (apart from cuts) are required to specify the integrand uniquely. Note that after integration, terms like the one drawn on the right of figure (53) vanish in dimensional regularization (power divergent terms are set to zero) and do not affect the final answer.

Similarly, as discussed in [108], the two-loop amplitude has poles at infinity for $n > 5$ where the degree of the pole grows with the number of external legs. In particular, for the

---

[10]In the context of dimensional regularization, such diagrams are power divergent and get set to zero. However, at the integrand level, we do not drop any such terms. For higher multiplicity, the question of non-cut-constructability has been raised previously in [108] even for two-loop amplitudes.

following integral topology,

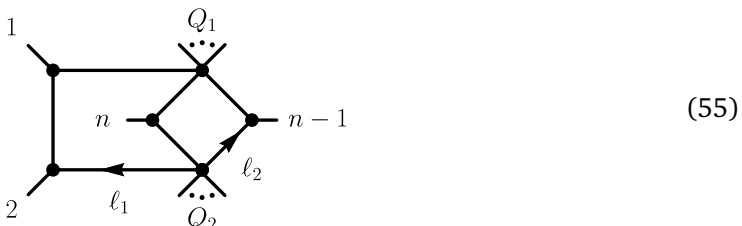

(55)

the degree of the pole on the maximal cut forces the numerator to take the form $N = (\ell_1 \cdot p)^{n-4}$. For $n \geq 12$ the ansatz would contain $(\ell_1 \cdot p)^8$ as well as four inverse propagators from the $\ell_2$-loop. We are again left with pure $d^4\ell_2$ integral without any propagators. Similarly for $n \geq 14$ we can collapse the $\ell_1$-loop.[11] Again, the two-loop $\mathcal{N} = 8$ amplitudes appear not cut-constructible for a sufficiently high number of points.

We would like to point out, that these higher-loop, higher-multiplicity statements go against the "no-triangle hypothesize" in $\mathcal{N} = 8$ SUGRA [53,109–111] that has been established by explicit one-loop calculations. At the level of the analytic structure, the absence of triangles at one loop goes hand-in-hand with the absence of poles at infinity [68,78,108]. As alluded to above, previous results of two of the authors [108] show that even starting at two loops, at sufficiently high multiplicity, the analytic structure of the amplitudes is such that higher poles at infinity are present, which requires the introduction of triangle integrand basis elements to match these poles.

**Unification of constraints**

The presence of the no-propagator integrands requires new constraints beyond unitarity cuts. One option is to also include constraints on the amplitude's dependence on external kinematics as a complement to constraints imposed by unitarity cuts. The BCFW scaling is a natural candidate, and we already saw the successful application of the external kinematic shifts in fixing the three-loop four-point amplitude. It is obvious that only the BCFW scaling *on* multiparticle unitarity cuts can not be enough, some basis integrals (as the one above) would directly vanish on these cuts, and further scaling would not impose any extra constraint. Therefore, the only possible resolution is the *simultaneous* scaling of both external and loop momenta at infinity, basically boosting the full amplitude to infinity. The scaling of multi-particle unitarity cuts as well as the large $t$ scaling under BCFW shifts would then just be special cases of this more general deformation. This also goes back to the study of the behavior at infinity of tree-level amplitudes under general shifts. These are very important questions and we leave them for future work.

## 5 New tree-level recursion relations

### 5.1 Helicity agnostic $(n-2)$-line shift

Motivated by our discussion, we can look more closely at shifts of graviton (pure GR, no susy) tree-level amplitudes, and explore the behavior under various shifts in $D = 4$. To reiterate the earlier discussion of the multi-particle unitarity cut; our parametrization of the on-shell loop momenta in Eq. (23) corresponds to a chiral shift of $(n-2)$ legs of the tree-level amplitude

---

[11]This counting is conservative and considers that $(\ell_1 \cdot p)^2 \sim \ell_1^2, \ell_2^2$. In the integral reduction we can often use relations such as $2(\ell_1 \cdot p) = (\ell_1 + p)^2 - \ell_1^2$ and the problem might appear even for lower $n$.

entering the cut

$$
\begin{array}{ccc}
\begin{array}{c} 2 \quad \ell_1 \\ \vdots \\ 1 \quad \ell_{L+1} \end{array}
& \longleftrightarrow &
\begin{array}{c} 2 \quad 3 \\ \vdots \\ 1 \quad n \end{array}
\end{array}
\tag{56}
$$

The particular behavior for $t \to \infty$ strongly depends on the distribution of helicities of the shifted legs. Just as in Eq. (23), we shift the "loop" momenta in an anti-chiral fashion $\widetilde{\lambda}_{\ell_i} \to \widetilde{\lambda}_{\ell_i} + t\, z_i \widetilde{\eta}$ subject to momentum conservation which imposes a constraint on the $z_i$. In order to find out, whether or not an amplitude is recursively constructible by such a shift, one should study the asymptotic behavior of the amplitude as $t \to \infty$. Here, we try to understand the scaling properties of tree-level graviton amplitudes by studying explicit "data". Several features stand out when looking at the large $t$ behavior in Tab. 2: generally, reading

Table 2: Scaling behavior of graviton tree-level amplitudes under the chiral deformation defined in Eq. (23) of $(n-2)$ external legs. The scaling is sorted according to the different helicity configurations of the amplitudes and shifted legs are denoted by $\widehat{\pm}$. The $L$-loop data corresponds to $(L+3)$-point tree-level amplitudes.

| two-loop data | | | | |
|---|---|---|---|---|
| $(--\widehat{+}\widehat{+}\widehat{+})\sim t^2$ | $(--\widehat{+}\widehat{+}\widehat{-})\sim t^1$ | | | |
| $(-+\widehat{+}\widehat{+}\widehat{-})\sim t^2$ | $(-+\widehat{+}\widehat{-}\widehat{-})\sim t^1$ | | | |
| $(++\widehat{+}\widehat{-}\widehat{-})\sim t^2$ | $(++\widehat{-}\widehat{-}\widehat{-})\sim t^{-7}$ | | | |
| three-loop data | | | | |
| $(--\widehat{+}\widehat{+}\widehat{+}\widehat{+})\sim t^3$ | $(--\widehat{+}\widehat{+}\widehat{+}\widehat{-})\sim t^2$ | $(--\widehat{+}\widehat{+}\widehat{-}\widehat{-})\sim t^0$ | | |
| $(-+\widehat{+}\widehat{+}\widehat{+}\widehat{-})\sim t^3$ | $(-+\widehat{+}\widehat{+}\widehat{-}\widehat{-})\sim t^2$ | $(-+\widehat{+}\widehat{-}\widehat{-}\widehat{-})\sim t^0$ | | |
| $(++\widehat{+}\widehat{+}\widehat{-}\widehat{-})\sim t^3$ | $(++\widehat{+}\widehat{-}\widehat{-}\widehat{-})\sim t^1$ | $(++\widehat{-}\widehat{-}\widehat{-}\widehat{-})\sim t^{-8}$ | | |
| four-loop data | | | | |
| $(--\widehat{+}\widehat{+}\widehat{+}\widehat{+}\widehat{+})\sim t^4$ | $(--\widehat{+}\widehat{+}\widehat{+}\widehat{+}\widehat{-})\sim t^3$ | $(--\widehat{+}\widehat{+}\widehat{+}\widehat{-}\widehat{-})\sim t^1$ | $(--\widehat{+}\widehat{+}\widehat{-}\widehat{-}\widehat{-})\sim t^{-1}$ | |
| $(-+\widehat{+}\widehat{+}\widehat{+}\widehat{+}\widehat{-})\sim t^4$ | $(-+\widehat{+}\widehat{+}\widehat{+}\widehat{-}\widehat{-})\sim t^3$ | $(-+\widehat{+}\widehat{+}\widehat{-}\widehat{-}\widehat{-})\sim t^1$ | $(-+\widehat{+}\widehat{-}\widehat{-}\widehat{-}\widehat{-})\sim t^{-1}$ | |
| $(++\widehat{+}\widehat{+}\widehat{+}\widehat{-}\widehat{-})\sim t^4$ | $(++\widehat{+}\widehat{+}\widehat{-}\widehat{-}\widehat{-})\sim t^2$ | $(++\widehat{+}\widehat{-}\widehat{-}\widehat{-}\widehat{-})\sim t^0$ | $(++\widehat{-}\widehat{-}\widehat{-}\widehat{-}\widehat{-})\sim t^{-9}$ | |
| five-loop data | | | | |
| $(--\widehat{+}\widehat{+}\widehat{+}\widehat{+}\widehat{+})\sim t^5$ | $(--\widehat{+}\widehat{+}\widehat{+}\widehat{+}\widehat{+}\widehat{-})\sim t^4$ | $(--\widehat{+}\widehat{+}\widehat{+}\widehat{+}\widehat{-}\widehat{-})\sim t^2$ | $(--\widehat{+}\widehat{+}\widehat{+}\widehat{-}\widehat{-}\widehat{-})\sim t^0$ | $(--\widehat{+}\widehat{+}\widehat{-}\widehat{-}\widehat{-}\widehat{-})\sim t^{-2}$ |
| $(-+\widehat{+}\widehat{+}\widehat{+}\widehat{+}\widehat{+}\widehat{-})\sim t^5$ | $(-+\widehat{+}\widehat{+}\widehat{+}\widehat{+}\widehat{-}\widehat{-})\sim t^4$ | $(-+\widehat{+}\widehat{+}\widehat{+}\widehat{-}\widehat{-}\widehat{-})\sim t^2$ | $(-+\widehat{+}\widehat{+}\widehat{-}\widehat{-}\widehat{-}\widehat{-})\sim t^0$ | $(-+\widehat{+}\widehat{-}\widehat{-}\widehat{-}\widehat{-}\widehat{-})\sim t^{-2}$ |
| $(++\widehat{+}\widehat{+}\widehat{+}\widehat{+}\widehat{-}\widehat{-})\sim t^5$ | $(++\widehat{+}\widehat{+}\widehat{+}\widehat{-}\widehat{-}\widehat{-})\sim t^3$ | $(++\widehat{+}\widehat{+}\widehat{-}\widehat{-}\widehat{-}\widehat{-})\sim t^1$ | $(++\widehat{+}\widehat{-}\widehat{-}\widehat{-}\widehat{-}\widehat{-})\sim t^{-1}$ | $(++\widehat{-}\widehat{-}\widehat{-}\widehat{-}\widehat{-}\widehat{-})\sim t^{-10}$ |

the table horizontally (i.e. for fixed helicities of the two unshifted legs), the more negative helicity gravitons we shift by our anti-holomorphic deformation, the better the large $t$ scaling. Generally, the anti-chiral deformation of $(n-2)$ legs leads to a large $t$ behavior of the amplitude that would not allow us to recursively reconstruct the answer due to the nontrivial contribution at infinity. It is also noteworthy that the large $t$ behavior cannot be simply predicted from little group scaling and the knowledge of the mass dimension of the amplitude alone. This is in contrast to a similar all-line shift analyzed previously [97] where the behavior can be predicted. As a simple example, consider the following shifted amplitude,

$$
\begin{array}{c}
2^- \quad \hat{3}^+ \\
\quad \hat{4}^+ \sim t^2 \;. \\
1^- \quad \hat{5}^+
\end{array}
\tag{57}
$$

To illustrate that the scaling of the tree amplitude does not simply follow from little group considerations, we can write down two example terms that have the same mass and little

group weights, but behave very differently under the chiral shift (23),

$$\text{term 1:} \quad \frac{\langle 12 \rangle^8 \langle 13 \rangle [13] \langle 23 \rangle [23]}{\langle 13 \rangle^2 \langle 14 \rangle \langle 15 \rangle \langle 23 \rangle^2 \langle 24 \rangle \langle 25 \rangle \langle 45 \rangle^2} \sim t^2$$
$$\text{term 2:} \quad \frac{([34][35][45])^2}{[12]^4} \sim t^6 . \tag{58}$$

One might wonder how various representations for gravity amplitudes compare with respect to their term-wise large $t$ limit. As a first representation, let us consider the local BCJ form written in terms of cubic diagrams. For the five-particle example of Eq. (57), explicit color-Jacobi satisfying numerators are known, see appendix D of [112]. Plugging these numerators into Eq. (4.6) of [113] and evaluating the individual terms on the shifted kinematics (23), we see that the terms in the BCJ representation scale like $t^4$. In order to reproduce the $t^2$ behavior of the full amplitude, cancellations between different terms are therefore necessary. This is rather interesting and demonstrates once more that local representations in gravity are not ideal to faithfully represent the true UV structure of the theory.

In contrast to the BCJ representation [113] which is not gauge (diffeomorphism) invariant term-by-term, we can also study the KLT representation [91–93] of the five-point gravity amplitude in Eq. (57). In KLT, one can expresses the full amplitude in terms of gauge invariant building blocks that, however, have spurious double poles and can schematically be written as[12] (see e.g. [114, 115])

$$\mathcal{M}_n^{\text{tree}} = \sum_{\sigma, \rho \in S_{n-3}} \mathcal{A}_n^{\text{tree}}(1, \sigma, n, n-1) S[\sigma|\rho] \mathcal{A}_n^{\text{tree}}(1, \rho, n-1, n), \tag{59}$$

where $S[\sigma|\rho]$ is the momentum dependent KLT kernel and the permutation sum is over the $(n-3)!$ permutations of legs $\{2, \ldots, n-2\}$. As written in Eq. (59), the three legs $\{1, n-1, n\}$ are special, but any other choice of three legs works equally well and as we will see in a moment might be preferable at times. To be more concrete, at five points, the KLT relation reads,

$$\mathcal{M}_5^{\text{tree}} = -s_{12}s_{13}\mathcal{A}_5(13245)\mathcal{A}_5(12354) - s_{13}(s_{12}+s_{23})\mathcal{A}_5(13245)\mathcal{A}_5(13254)$$
$$- s_{12}s_{13}\mathcal{A}_5(12345)\mathcal{A}_5(13254) - s_{12}(s_{13}+s_{23})\mathcal{A}_5(12345)\mathcal{A}_5(12354). \tag{60}$$

From the UV perspective, somewhat surprisingly, the KLT representation has extremely desirable properties. In fact, "term 1" in eq. (58) is the worst behaved term in the KLT form of the amplitude (60), yet scales much better at large $t$ as "term 2" in eq. (58) or the BCJ pieces. In particular, one can check that the $s_{13}s_{23}\mathcal{A}_5(13245)\mathcal{A}_5(13254)$ term in (60) has the same large $t$ scaling as the amplitude in Fig. 57 itself. This somewhat interesting observation empirically extends to all other cases we have studied. From the point of view of KLT, the behavior of the gravity amplitudes under the chiral shift is therefore inherited from the large $t$ behavior of the Yang-Mills tree amplitudes. This is even more evident when one chooses the two unshifted legs as special in the KLT formula and realizes that the KLT kernel scales uniformly at large $t$ like $t^{n-3}$, for $n \geq 5$.

For special helicity configurations where only MHV amplitudes contribute, the anti-chiral shift $\widetilde{\lambda}_{\ell_i} \to \widetilde{\lambda}_{\ell_i} + t z_i \widetilde{\eta}$ does not affect the Yang-Mills trees at all and the UV scaling comes entirely from the KLT kernel (in agreement with the scaling data in Tab. 2). Likewise, if only $\overline{\text{MHV}}$ amplitudes are involved and the two positive helicity gravitons are taken to be $1^+$ and $2^+$, with all other gravitons having negative helicity, one can simply count the scaling of the building blocks in (a relabeled $n \leftrightarrow 2$ version of) Eq. (59): $S[\sigma|\rho] \sim t^{n-3}$, $\mathcal{A}_n^{\text{tree}}(1^+, \sigma, 2^+, n-1) \sim 1/t^n$

---

[12]In all previous cases, we collectively denoted amplitudes by $\mathcal{A}$ representing either gravity or Yang-Mills depending on the context. Here we explicitly distinguish GR ($\mathcal{M}$) from YM ($\mathcal{A}$).

and $\mathcal{A}_n^{\text{tree}}(1^+, \rho, n-1, 2^+) \sim 1/t^{n-1}$ which gives the observed $1/t^{n+2}$ of this shift-sector in Tab. 2. In the second Yang-Mills factor, one power of $t$ cancels because the two positive helicity particles are adjacent in the anti-Parke-Taylor factor and $[12]$ does not scale with $t$. For other helicity components, the analysis is much more involved but boils down to analyzing Yang-Mills tree amplitudes.

In conclusion, it seems to be the case that the KLT representation of gravity tree-amplitudes, despite obscuring some properties (such as locality), manifests the UV scaling behavior term-by-term. It would be interesting to study this in more generality—including in $D$ dimensions.

## 5.2 Same helicity $m$-line shift

Even though gravity amplitudes already show an improved large $t$ behavior under the anti-holomorphic shift (23), the fall off at infinity is generically still not good enough in order to recursively construct the amplitudes. Here we study a variant of the chiral shift defined in Eq. (23)

$$\widetilde{\lambda}_j \mapsto \widetilde{\lambda}_j + t \, z_j \, \widetilde{\eta} \quad \text{for} \quad j \subset \{1, \ldots, n\} \quad \text{subject to} \quad \sum_{j \subset \{1,\ldots,n\}} z_j \lambda_j = 0 \,, \tag{61}$$

but now allow ourselves to shift *any* number of legs, not just $(n-2)$. The special case of a $k$-line shift (where all $k$ negative helicity particles of an $N^{k-2}$MHV amplitude are deformed) was initially studied in [116] in order to derive the CSW rules [117] in gauge theory. Applied to NMHV amplitudes in gravity, [97] concluded that the CSW recursion relations break down at $n = 12$. In particular, the behavior of the $n$-particle NMHV amplitude under the shift (61) is

$$\mathcal{M}_n^{\text{NMHV}}(t) \sim \frac{1}{t^{12-n}} \,. \tag{62}$$

It is obvious from the data in Tab. 2 that in order to get a good large $t$ behavior we can only shift $(-)$ helicity gravitons. The dependence of the amplitude on the $(+)$ helicity gravitons requires extra $\widetilde{\lambda}$-dependent factors in the numerator to get the correct little group weight. Shifting these gravitons then deteriorates or spoils the large $t$ behavior.

Consider $n$-point $N^{k-2}$MHV amplitudes with $k$ negative helicity gravitons. We shift $m \leq k$ of these legs via the deformation in Eq. (61). Based on experimental evidence up to eight-points, we conclude that the large $t$ behavior of this $m$-line shift is

$$\mathcal{M}_n^{N^{k-2}\text{MHV}}(t) \sim \frac{1}{t^{6+m-(n-k)}} \,. \tag{63}$$

The more general scaling (63) is in agreement with the 3-line shift scaling of Eq. (62), as seen by setting $m = 3$, and the least favorable value $k = 3$. Note that $k = 2$ is not covered by this analysis because we need at least three negative helicity gravitons to perform the shift (61).

The $m$-line shift provides more flexibility, and shifting a sufficient number of external legs one can always achieve constructibility ($\mathcal{M}_n^{N^{k-2}\text{MHV}}(t)$ falls off at least like $1/t$ as $t \to \infty$). In particular, for $k > \frac{n}{2}$ we use the chiral shift (61) and the exponent is positive for $m > (n-k)-6$, i.e. if we shift more than $(n-k)-6$ negative helicity gravitons. In the extreme case when all $(-)$ gravitons are shifted, we have $m = k$ and the amplitude scales as

$$\mathcal{M}_n^{N^{k-2}\text{MHV}}(t) \sim \frac{1}{t^{6+2k-n}} < \frac{1}{t^6} \,, \qquad \text{for } k > \frac{n}{2} \,. \tag{64}$$

If $k < \frac{n}{2}$ we can use the holomorphic chiral shift

$$\lambda_j \to \lambda_j + t \, z_j \, \eta \quad \text{for} \quad j \subset \{1, \ldots, n\} \quad \text{subject to} \quad \sum_{j \subset \{1,\ldots,n\}} z_j \widetilde{\lambda}_j = 0 \,, \tag{65}$$

and repeat exactly the same exercise as before. For $k = \frac{n}{2}$ both shifts are equivalent, giving a large $t$ scaling behavior of $\mathcal{M}_n^{\mathrm{N}^{\frac{n}{2}-2}\mathrm{MHV}}(t) \sim \frac{1}{t^6}$ for the maximal $m = k = \frac{n}{2}$ shift. In summary, we can always choose a shift such that the behavior at infinity is at least $\frac{1}{t^6}$.

**Bonus relations**

If the deformed amplitude $\mathcal{M}_n(t)$ falls off at large $t$ at least like $1/t$, we can consider a Cauchy residue theorem of the shifted amplitude starting from a little contour around infinite $t$, $\mathcal{C}_\infty$,

$$\oint_{\mathcal{C}_\infty} \frac{dt}{t} \mathcal{M}_n(t) = 0 \quad \leftrightarrow \quad \mathcal{M}_n(t=0) = -\sum_{\substack{i \in \text{poles} \\ \text{of } \mathcal{M}_n(t)}} \frac{1}{t_i} \mathrm{Res}_i \mathcal{M}_n(t=t_i). \tag{66}$$

The residues at the poles $t = t_i$, $\mathrm{Res}_i \mathcal{M}_n(t=t_i)$, are then calculated as products of shifted lower point amplitudes as usual. This is obviously not the most economic shift as the usual BCFW shift scales like $\frac{1}{t^2}$ for any helicity configuration (except one) and therefore is sufficient to reconstruct any tree-level amplitude recursively. However, it is still quite interesting to see that there is another set of shifts with even milder behavior at infinity. This improved behavior at infinity then leads to a wider range of bonus relations (see [51,52,95,96] and the discussion in Section 3.3) of the form

$$\mathcal{M}_n(t) \sim \frac{1}{t^r} \quad \text{for} \quad t \to \infty \leftrightarrow 0 = \oint_{\mathcal{C}_\infty} dt\, t^{r-2} \mathcal{M}_n(t) = \sum_{\substack{i \in \text{poles} \\ \text{of } \mathcal{M}_n(t)}} t_i^{r-2} \mathrm{Res}_i \mathcal{M}_n(t=t_i). \tag{67}$$

It would be interesting to investigate whether imposing the behavior at infinity is enough to fix the tree amplitudes uniquely. In [118] it was shown that the $z^{-1}$ behavior of Yang-Mills tree amplitudes under BCFW shifts is enough to specify them uniquely at leading order in the soft expansion. More recently, similar UV scaling constraints were used to uniquely fix tree-level amplitudes in a variety of effective field theories [119]. The gravity shifts explored above, in conjunction with the BCFW shift, should provide more flexibility and be part of a larger story of how graviton tree-level amplitudes behave at infinity.

The large $t$ behavior (21) of the multi-particle unitarity cut (11) is inherited from the large $t$ scaling of gravity tree-level amplitudes. We saw that this behavior can be understood from KLT as a consequence of the large $t$ scaling of Yang-Mills amplitudes, and is not a property of individual BCJ terms. This suggests that Yang-Mills amplitudes are responsible for the unexpected behavior of gravity cuts. On the other hand, the improved behavior of the gravity cut was special to $D = 4$, while KLT works in any spacetime dimension. Furthermore, the same cut in Yang-Mills did not improve in the $D = 4$ limit, so that the improved scaling of the gravity cut (11) can not be explained (solely) by gauge invariance or color-kinematics duality and some new ingredient is required.

## 6 Conclusion and Outlook

This work is part of a larger program to understand the structure of scattering amplitudes at large momenta. Concretely, the goal is to understand what is "unitarity at infinity" and how amplitudes behave in this limit. Perturbative unitarity at finite loop momenta describes the factorization of loop integrands on poles, closely related to discontinuities of final amplitudes on branch cuts. No analogous statement is known about singularities located at infinite loop momenta. In this paper, we gathered further evidence that (cuts of) gravity loop integrands

possess unexpected properties at large loop momenta. We have investigated this phenomenon for multi-particle unitarity cuts through seven loops and saw that an improved scaling behavior is present in $D = 4$ thanks to the vanishing of Gram determinants. This suggests that gravity amplitudes have more structure in $D = 4$ than in general $D$, but further details remain to be understood.

The scaling properties of the integrand at infinity should also be reflected in the structure of final amplitudes in $D = 4$. As stated in the introduction, there exists an allowed counterterm, $D^8 R^4$, in $\mathcal{N} = 8$ supergravity consistent with all known symmetries of the theory (supersymmetry and duality symmetry), relevant for a potential UV divergence at seven loops in $D = 4$. The recent computation by Bern et al. demonstrates the presence of this counterterm in $D_c = 24/5$. However, counterterms that are allowed and present in higher dimensions, are not necessarily present in $D = 4$. A simple example is $R^4$ in $\mathcal{N} = 4$ sugra at 1-loop which is the relevant counterterm for the $D = 8$ divergence, but it does not show up in $D = 4$ at three loops [17, 23, 24]. We do not claim that this has to be the fate of $D^8 R^4$ in $D = 4$, but our on-shell analysis suggests that $D = 4$ is indeed very special from the on-shell perspective. Only in $D = 4$ we see certain integrand level UV cancellations which could hint at enhanced cancellations after integration.

The behavior of loop integrands on unitarity cuts is directly tied to the behavior of tree-level amplitudes, and our multi-line shift provides additional evidence of non-trivial scaling at infinity similar to the BCFW shift. In the last part of the paper we used the scaling properties of cuts as homogeneous constraints to fully fix the loop integrand. Our analysis suggests that a more general framework to probe the behavior of amplitudes at infinity is to simultaneously shift both (cut) loop momenta as well as external momenta, and scale them to infinity at the same time. Our ability to fix gravity amplitudes in certain examples using only homogeneous constraints suggests a possible geometric interpretation similar to the planar $\mathcal{N} = 4$ SYM case. Even if a geometric formulation for gravity is speculative, our analysis teaches us some important lessons about gravity, namely that the improved behavior at infinity in fact controls the full amplitude. We also used the same shift to construct the tree-level amplitudes using recursion relations.

It is important to stress that the cancelations at infinity uncovered in this paper do not seem to be a consequence of gauge invariance or supersymmetry, as individual terms in the integrand basis with their coefficients are both supersymmetric and gauge invariant. Furthermore, similar cancelations are present in any two-derivative theory of gravity, supersymmetric or not. The reason we focus on $\mathcal{N} = 8$ supergravity is because of the availability of explicit integrand data [87–89, 120] to compare against, while for pure GR the results are limited. Also, the uniqueness construction using homogeneous data likely works only for the maximally supersymmetric case, whereas for pure GR we have to supplement additional information.

For planar $\mathcal{N} = 4$ SYM in $D = 4$, the (complete) absence of poles at infinity is a consequence of dual conformal symmetry [85, 86]. The same property was conjectured to be true for the full (non-planar) $\mathcal{N} = 4$ SYM theory suggesting there is a hidden symmetry in the full theory too. First steps in this direction have been pursued in [49, 78], see also interesting related work [81, 121, 122].

We have seen earlier that gravity loop integrands do have poles at infinity, as demonstrated on maximal cuts. Therefore, no analogue of dual conformal symmetry can be present. On the other hand, the poles at infinity surprisingly cancel in certain directions when approaching infinity in $D = 4$. Our observations fall into the same category as the large $z$ behavior of amplitudes under BCFW shifts, and provide further evidence that something is missing in our understanding of gravity amplitudes. Following the traditional logic that the properties of the S-matrix are a consequence of symmetries, it is suggestive that this phenomenon is indeed caused by some yet-to-be found symmetry or some novel property of general relativity.

# Acknowledgements

We are grateful to Zvi Bern, Henriette Elvang, and Chia-Hsien Shen for enlightening discussions and comments on the manuscript. A.E. and J.P.-M. thank the Mani L. Bhaumik Institute for generous support. J.P.-M. also thanks SLAC for hospitality. J.P.-M. is supported by the U.S. Department of State through a Fulbright Scholarship. E.H. is grateful to the Mani L. Bhaumik Institute for Theoretical Physics at UCLA, for hospitality during various stages of this project, and the Aspen Center for Physics, which is supported by National Science Foundation grant PHY-1607611. The research of A.E. is in part supported by the Knut and Alice Wallenberg Foundation under KAW 2018.0116, *From Scattering Amplitudes to Gravitational Waves*. The work of E.H. is supported by the U.S. Department of Energy (DOE) under contract DE-AC02-76SF00515. The research of J.T. is supported in part by U.S. Department of Energy grant DE-SC0009999 and by the funds of University of California.

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
