# Peer review of "Gravity loop integrands from the ultraviolet"

_SciPost Physics, doi:SciPost Phys. 10, 016 (2021)_

## Round 1 · Referee Report · Anonymous (Referee 1) · 2020-10-11

Strengths

1- Very clear discussion of the results 2- Paper gives important insight into the structure of gravity amplitudes

Weaknesses

none

Report

The paper discusses UV properties of gravity amplitudes, both in pure gravity and and supersymmetric theories. It proposes an unexpected scaling behaviour of the amplitudes for large loop momenta in 4 dimensions (which is absent for YM). Understanding the UV properties of gravity amplitudes in 4 dimensions is important in order to uncover possible cancellations which may lead to an improved UV behaviour of gravity amplitudes. This in turn may lead to new ways to prove, or disprove, the UV finiteness of N=8 SUGRA in 4 dimensions. The paper proposes an interesting new direction to understand the UV structure of 4-dimensional gravity theories.

The paper is well written, and the results are clearly presented and discussed. I recommend the paper for publication, but I suggest that the authors address the points raised below.

Requested changes

1- At the very bottom of page 13, the authors say that they "get e.g. $t^4$ for GR and $1/t^{L−2}$ for YM". While I can see the $1/t^{L−2}$ scaling in Figure 2, I seem to see a scaling of $t^3$ for GR. Is this a typo? The authors should clarify this point.

2- Figure 2 could benefit from having also the scaling of the $D$-dimensional amplitudes on the same plot (e.g., as a thin line). This would convey more clearly the message of the difference between the 4- and $D$-dimensional cases.

---

## Round 1 · Referee Report · Anonymous (Referee 2) · 2020-10-15

Report

The manuscript "Gravity loop integrands from the ultraviolet" studies the
behaviour of the integrand of $\mathcal{N} = 8$ loop amplitudes at large momentum.

There are three technical points, which are of interest to the community.
The new results of the authors are:
(i) four-dimensional unitarity cuts have a better ultraviolet behaviour than expected.
(ii) a reconstruction of the loop integrand from the improved ultraviolet scaling and the vanishing
at particular points in momentum space
(iii) the discussion of new multi-line shifts for BCFW-type recursion relations.

These results, albeit technical in nature, are helpful for the community and worth publishing.

The driving force behind these results is the desire of the authors to understand the "raison d'etre"
for the improved ultraviolet behaviour.
The authors openly admit in their manuscript that so far they have not found a good explanation.
At various places the authors hint at speculations, without giving sufficient information to the readers.
This makes it difficult for the readers to understand what the authors have in mind.
It would not degrade the manuscript, if these passages are left out.

There are a few smaller points, where clarifications could be helpful:

Page 6: "minimal power counting of an integral": The authors could define this term.
The authors could also elaborate why the maximal cuts determine this, the readers might get lost at this point otherwise.

Page 27, section 4.3: " ... it shows that the $\mathcal{N} = 8$ supergravity loop integrand can be fully fixed
using only homogeneous constraints at infinity."
In general or just for the examples discussed above?

Page 31, two lines after eq.(5.5): Does "term 1" refer to the first term in eq.(5.5) or to eq.(5.3)?

---

## Round 2 · Author Response

We thank the referees for useful suggestions that have improved the general presentation of the paper.
We believe these changes address all the points raised by the referees and hope these improvements are sufficient to allow publication.

---

## Round 2 · List of Changes

We have implemented the following changes:

  1. The typographical error in the scaling for general relativity at the bottom of page 13 pointed out in Report 1 has been corrected.
  2. Thin lines have been added to Figure 2 explaining the scaling of D-dimensional unitarity cuts.
  3. Footnote 6 now contains a definition of “minimal power-counting”.
  4. A parenthetical remark has been added in section 4.3 clarifying which integrands in N=8 can be fixed via homogeneous constraints.
  5. Eq. (5.3) has been modified to clarify references in the text to “term 1” and “term 2”.
  6. The more speculative comments about the implications of the results of this work have been removed from the main text and adapted for the conclusions.

---

## Editorial Decision

published